# A two-stage hybrid NSGA-III with BWO for path planning and task allocation in agricultural land preparation

Manxian Yang[1,2,3], Yanhong Chen◉[1,2,3]*, Yongke Li[1,2,3]*, Taihong Zhang[1,2,3], Tianlun Wu◉[4,5]

1 College of Computer and Information Engineering, Xinjiang Agricultural University, Urumqi, P.R. China,
2 Engineering Research Center of Intelligent Agriculture Ministry of Education, Urumqi, P.R. China,
3 Xinjiang Agricultural Informatization Engineering Technology Research Center, Urumqi, P.R. China,
4 College of Mechanical and Electrical Engineering, Xinjiang Agricultural University, Urumqi, P.R. China,
5 Xinjiang Key Laboratory of Intelligent Agricultural Equipment, Urumqi, P.R. China

* cyh@xjau.edu.cn (YC); lyk@xjau.edu.cn (YL)

**Data Availability Statement:** Our data is openly available in the following github repository: https://github.com/yangmanxian/BNSGA-III_data.

**Funding:** This study was supported by the Central Government Guiding Local Science and Technology Development Special Fund Project

## Abstract

Automated large-scale farmland preparation operations face significant challenges related to path planning efficiency and uniformity in resource allocation. To improve agricultural production efficiency and reduce operational costs, an enhanced method for planning land preparation paths is proposed. In the initial stage, unmanned aerial vehicles (UAVs) are employed to collect data from the field, which is then used to construct accurate farm models. For single-field operations, a path planning approach is developed that minimizes energy consumption. The approach combines the selection of optimal operational angles with the implementation of efficient turning strategies, aiming to achieve full coverage. In addressing the issue of scheduling multiple machines across multiple fields, a two-stage optimization method, referred to as the BNSGA-III algorithm, is introduced. This algorithm integrates the NSGA-III algorithm with Beluga Whale Optimization (BWO), adaptive parameter adjustment, and Adaptive Inversion Crossover (AIC). The proposed method tackles the inherent complexity of agricultural environments, balancing operational efficiency and resource allocation through multi-objective optimization. Experimental results demonstrate that, compared to random operation directions, the proposed method reduces the path length by 1.9% to 3.1%, decreases the turning frequency by 19.5% to 24.0%, and improves coverage by 1.0% to 1.4%. In the context of multi-machine scheduling, the BNSGA-III algorithm outperforms the NSGA-II, NSGA-III, and MOEA/D algorithms, achieving improvements in total travel distance (12.3% to 34.4%), path balance (60.9% to 66.2%), and workload distribution (78.7% to 92.9%). Further evaluation shows that BNSGA-III excels in key metrics such as convergence (IGD), solution quality (HV), and diversity (Spread), thereby confirming its superiority in solution quality, convergence, and diversity. The findings of this study provide strong support for the advancement of intelligent agriculture.

"Application and Promotion of Key Technologies for Integrated Water and Fertilizer Management in Desert Wheat" (ZYYD2024CG19); the Ministry of Science and Technology Innovation 2030 Major Project on "New Generation Artificial Intelligence"—"Integrated Application and Demonstration of Key Technologies for Smart Farms" (2022ZD0115805); and the Xinjiang Uygur Autonomous Region Major Science and Technology Special Project "Research on Intelligent Control System for Integrated Water and Fertilizer Management in Farmland" (2022A02011-3).

**Competing interests:** The authors have declared that no competing interests exist.

# 1. Introduction

## 1.1 Research motivation and background

The evolution of modern agriculture towards large-scale, precision operations has positioned agricultural machinery automation as a critical technological advancement for enhancing production efficiency. The implementation of autonomous agricultural machinery has the potential to significantly reduce labor costs by 40% to 60%, as evidenced by studies [1, 2]. In the context of large-scale farm management, land preparation is a fundamental agricultural process, the quality of which directly impacts subsequent farming activities and crop development [3]. The utilization of scientific land preparation methods has been shown to confer numerous benefits, including increased crop yields, improved soil structure, and reduced weed proliferation [4].

As farm sizes increase, the use of multi-machine collaborative operations has become a crucial strategy for improving operational efficiency, as illustrated in Fig 1. However, several significant challenges must be addressed in practical applications. Complex terrain and irregular field boundaries limit the effectiveness of traditional path planning methods [5]. In multi-machine collaborative operations, unbalanced task allocation leads to inefficient resource

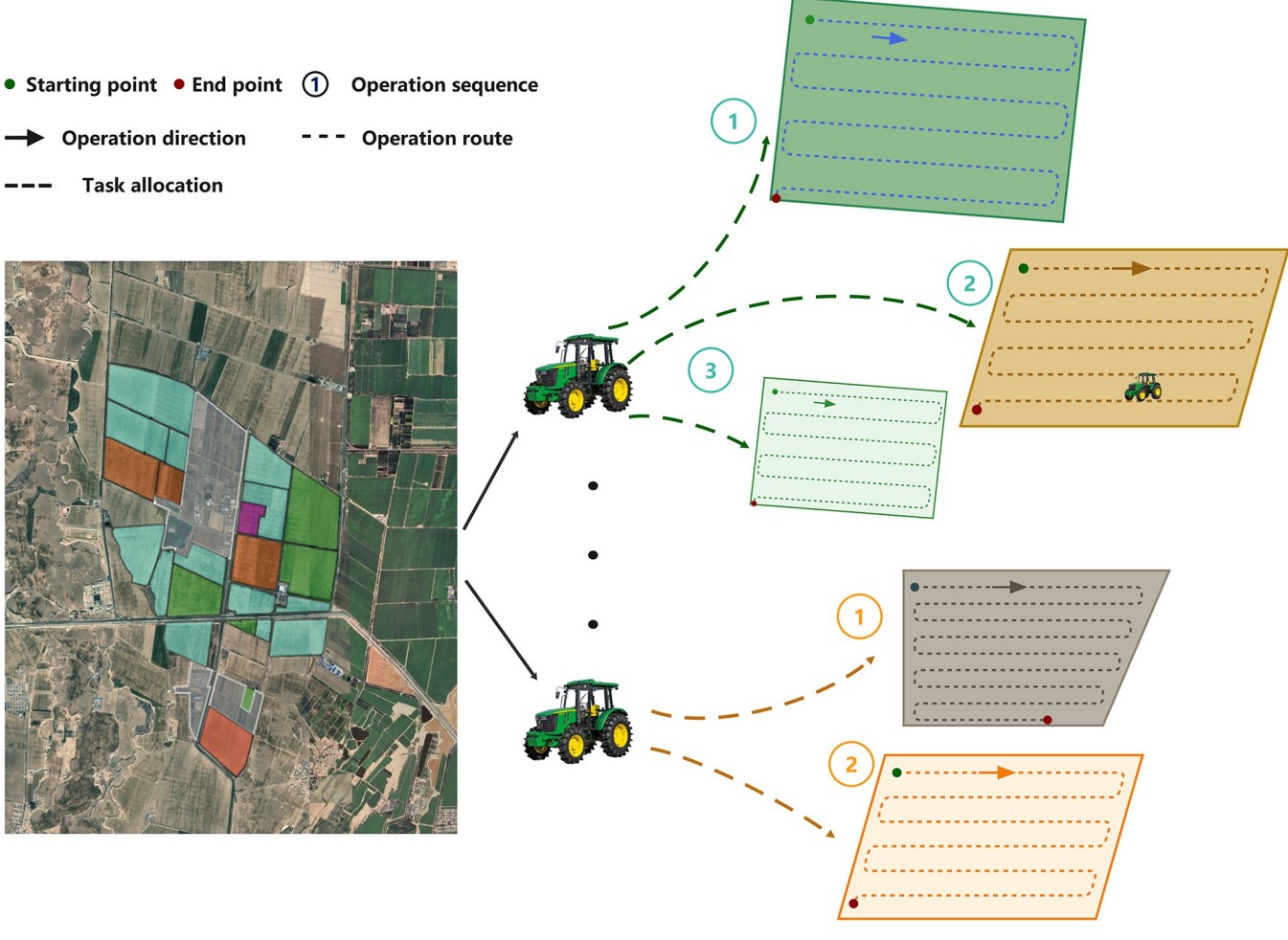

**Fig 1. Farm land preparation operation scheduling plan.**

utilization [6]. Moreover, insufficient consideration of actual road network constraints during inter-field machinery transfers results in additional energy consumption [7].

These technological limitations significantly hinder the advancement of farm automation. Path planning is a fundamental technology in automated land preparation, encompassing both the optimization of individual field coverage paths and the scheduling of collaborative operations across multiple fields. The former focuses on optimizing operational paths, while the latter addresses task allocation in the context of multi-machine cooperation. Resolving these technical challenges has the potential to substantially enhance farm operational efficiency and reduce production costs.

## 1.2 Related work

### 1.2.1 Single-field coverage path planning.
The majority of research conducted on single-field coverage path planning has concentrated on optimizing the operations of agricultural machinery to achieve comprehensive area coverage. In recent years, several significant approaches have emerged. Govindaraju et al. [8] proposed an optimized offline multi-robot coverage algorithm for paddy fields, enhancing weeding efficiency by reducing turns while ensuring comprehensive coverage. Nilsson et al. [9] developed a path planning method based on an improved artificial bee colony algorithm, designating full plot coverage as the primary objective. Building upon this work, Wu et al. [10] further refined the artificial bee colony algorithm to optimize path length and operational time while maintaining full coverage, thereby reducing energy consumption. While traditional path planning methods demonstrate satisfactory performance in simple environments, they often prove inadequate in complex terrain conditions. To address this limitation, Yan et al. [11] integrated chaos mapping and simulated annealing mechanisms into an improved ACO algorithm, effectively generating optimal trajectories and resolving obstacle avoidance challenges in complex environments.

### 1.2.2 Multi-field path planning.
The majority of research conducted on multi-field path planning has focused on optimizing the allocation of tasks and operational sequences for multiple agricultural machines. The majority of current studies employ intelligent algorithms based on traversal indicators to determine optimal field sequences. Nevertheless, the majority of existing research is concentrated on UAVs and robots, with relatively fewer studies dedicated to agricultural machinery [12–14]. In the field of task allocation optimization, a number of innovative approaches have been developed. Ali et al. [15] proposed a hybrid optimization algorithm, combining PSO and MAS, for the control of multi-UAV formations and the planning of paths. Shen et al. [16] developed an energy consumption model integrated with genetic algorithms to optimize farmland connection paths, thereby reducing energy expenditure. Similarly, A. et al. [17] put forth a multi-robot task allocation framework based on dynamic distributed particle swarm optimization, which enhanced operational efficiency. Venkatesh et al. [18] employed an artificial bee colony algorithm to address the single-depot multi-transportation-service-path (MTSP) problem, with the objective of optimizing both the total and maximum travel distances. Li et al. [6] enhanced task allocation for harvesters and grain transport vehicles through an improved ant colony algorithm integrated with NSGA-III, although the potential impact of variations in machinery numbers was not considered. Cao et al. [19] further enhanced the ant colony algorithm by incorporating agricultural plot data, thereby reducing the costs associated with harvesting and grain transportation routes. Furthermore, Kang et al. [20] introduced a multi-objective teaching-learning optimization algorithm to optimize task allocation for multiple weeding robots, with the objective of minimizing maximum completion time and remaining herbicide quantities.

In the realm of UAV task allocation. Huang et al. [21] employed sequential quadratic programming to optimize flight time. Yan et al. [22] implemented deep reinforcement learning for UAV trajectory planning and resource allocation, enhancing system performance through the integration of long short-term memory networks to accommodate dynamic task characteristics. Wang et al. [23] devised a multi-objective ant colony system that resulted in a reduction in operational time through the optimization of robot task allocation. Chen et al. [24] developed a heuristic algorithm based on ant colony systems, which improved task completion efficiency through area allocation and path optimization.

**1.2.3 Limitations of existing research.**   Despite the progress made in the development of individual path planning and task allocation methodologies, several significant limitations remain prevalent in the current research landscape. First, there is a lack of systematic optimization approaches for large-scale farm operations, with a notable absence of coherent optimization between single-field and multi-field scenarios. Second, the practical constraints of road networks and field entrance/exit restrictions are not adequately considered, which affects the feasibility of the proposed solutions. Third, the methods employed for task allocation in multi-machine collaborative operations are overly simplistic, making it difficult to achieve an effective balance between multiple optimization objectives.

## 1.3 Contributions

This study investigates the planning of paths for autonomous agricultural machinery and the allocation of tasks to such machinery, proposing a systematic multi-level optimization framework. The primary contributions are as follows:

(1) The development of a high-precision farm mapping system that incorporates field entrance and exit restrictions, as well as actual road network constraints. This system provides map services and algorithmic decision support.

(2) The implementation of a coverage path planning strategy for land preparation operations, which optimizes both coverage rate and turning frequency.

(3) The design of the BNSGA-III algorithm for scheduling multi-machine collaborative operations, featuring the following characteristics: A two-stage optimization strategy tailored to the specific characteristics of agricultural fields. This strategy integrates the BWO algorithm with adaptive parameter adjustment. Additionally, the introduction of AIC operations enhances population diversity.

## 1.4 Paper organization

The remainder of this paper is organized as follows: Section 2 describes the problem and presents a framework for its solution. Section 3 provides a model of farmland environments. Section 4 details the planning of operational paths for a single field, including the selection of driving patterns, the determination of operational directions, and the identification of appropriate turning methods. Section 5 proposes a collaborative operation scheduling method for multiple machines and fields, outlining enhancements to the BNSGA-III algorithm. Section 6 validates the proposed method through single-field and multi-machine scheduling experiments. Finally, Section 7 summarizes the paper and outlines potential avenues for future research.

# 2. Problem description and its solution

## 2.1 Problem description and modelling

In the context of large-scale agricultural operations, land preparation requires the use of tractors, which must depart from a central depot, perform operations across multiple fields in a

sequential manner, and subsequently return to the depot. To minimize energy consumption costs, it is essential to determine the optimal path that traverses all fields. Given the extensive acreage of farmland, it is typical for multiple tractors to operate simultaneously, transforming this into a Multiple Traveling Salesman Problem (MTSP) [25]. This framework enables synchronized path planning for multiple tractors, optimizing the allocation of machinery and tasks while avoiding operational redundancy and considering inter-field traversal sequences to minimize overall energy consumption.

Consider a large-scale farm F, comprising a set of n fields that require preparation (denoted by F = {f₁, f₂,. . ., fₙ}). Additionally, assume that the farm has m agricultural machines (denoted by M = {1,. . ., m}). The primary objective is to achieve efficient automated land preparation across all fields. This complex problem can be decomposed into three key sub-problems:

(1) Farmland Environment Modeling

To ensure path planning accuracy, a farm environment model is constructed using GPS and GIS technologies. Each field $f_i$ is defined as follows: $f_i = \{B_i, E_i, R_i\}, i = 1, 2, \ldots, n$

Where: $B_i = \{(x_1, y_1), (x_2, y_2), \ldots, (x_k, y_k)\}$ represents the boundary point set with a precision of $\leq \pm 5$ cm. $E_i = \{(ex_1, ey_1), (ex_2, ey_2), \ldots, (ex, ey)\}$ denotes the set of entrance/exit locations. $R_i$ represents the connecting road network.

(2) Single-field Coverage Path Planning

For each field $f_i$, a coverage path satisfying operational requirements must be generated. The optimization objective is:

$$\begin{cases} \min f(x) = (1 - \beta)L_{path} + \beta N_{turn} \\ \quad \text{Subject to :} \\ \quad C_{rate} \geq C_{min} \\ \quad \theta \in [0°, 180°] \\ \quad v \in [v_{min}, v_{max}] \\ \quad r \geq r_{min} \end{cases}, \quad (1)$$

Where: $L_{path}$ represents the total path length (m). $N_{turn}$ denotes the number of turns. $\beta \in [0,1]$ is an adaptive weight coefficient. $C_{rate}$ represents coverage rate. v is the operating speed. $v_{min}$ = 0.5m/s is the minimum speed. $v_{max}$ = 2m/s is the maximum speed. r is the turning radius. $r_{min}$ = 3.5m is the minimum turning radius.

(3) Multi-machine Multi-field Task Allocation

To address the path planning problem for multiple machine operators navigating between various operational points in the context of multi-machine, multi-field collaborative scheduling, a set of objective functions is designed based on the characteristics of the MTSP. Given the significant correlation between energy consumption, task completion time, and workload, a multi-objective optimization model is established for mmm machines and n fields, with three key objectives:

**Minimize the total collaborative travel distance**, ensuring that each machine travels the shortest possible distance to the next field after completing operations in a current field. This approach reduces both energy consumption and the time spent on inter-field transfers.

**Balance the paths** as much as possible, avoiding situations where some paths are excessively long while others are disproportionately short.

**Distribute the workload rationally** across each agricultural machine, ensuring uniformity in task distribution and preventing individual machines from being overloaded.

The optimization algorithm simultaneously considers these three objectives, thereby achieving superior performance in terms of solution quality and equilibrium. The following

objective functions are used for fitness evaluation:

$$
\begin{cases}
f1 = \sum_{k=1}^{m} \sum_{i=1}^{n} \sum_{j=1}^{n} C_{ij} \cdot X_{ijk} \\
f2 = \sqrt{\dfrac{1}{m} \sum_{k=1}^{m} (L_k - \mu)^2} \\
f3 = max_k \omega_k - min_k \omega_k
\end{cases}
\tag{2}
$$

where: $L_k = \sum_{i=1}^{n} \sum_{j=1}^{n} c_{ij} x_{ijk}$ is the length of the kth path. $\mu = \frac{1}{m} \sum_{k=1}^{m} L_k$ is the average path length. $W_k = \sum_{i=1}^{n} \sum_{j=1}^{n} w_{ij} x_{ijk}$ is the workload of the kth machine.

The multi-machine scheduling problem is subject to the following constraints to ensure operational feasibility:

Single Visit Constraint: Each operation point must be visited exactly once by one machine.

$$
\sum_{k=1}^{n} \sum_{i=1}^{n} x_{ijk} = 1, \forall j \in N
\tag{3}
$$

Flow Balance Constraint: For each machine and each point, the number of incoming routes equals the number of outgoing routes.

$$
\sum_{j=1}^{n} x_{ijk} = \sum_{j=1}^{n} x_{jik}, \forall j \in N, \forall k \in M
\tag{4}
$$

Workload Constraint: The total workload assigned to each machine cannot exceed the maximum limit.

$$
\sum_{i=1}^{n} \sum_{j=1}^{n} w_{ij} x_{ijk} \leq W_{max}, \forall k \in M
\tag{5}
$$

Total Time Constraint: The total operation time cannot exceed the specified time limit.

$$
\sum_{k=1}^{m} \sum_{i=1}^{n} \sum_{j=1}^{n} t_{ij} x_{ijk} \leq T_{total}
\tag{6}
$$

Decision Variable Constraints: Binary decision variables for route selection.

$$
x_{ijk} \in \{0, 1\}, \forall i, j \in N, \forall K \in M
\tag{7}
$$

Depot Constraints: Each machine must start from and return to the depot (denoted as point 0).

$$
\begin{cases}
\sum_{j=1}^{n} x_{0jm} = 1, \forall m \in M \\
\sum_{i=1}^{n} x_{i0m} = 1, \forall m \in M
\end{cases}
\tag{8}
$$

Parameter description:

$c_i$ is the distance from point i to j. $w_i$ is the workload from point i to j. $t_i$ is the operation time from point i to j. $W_{ax}$ is the maximum workload limit per machine. $T_{total}$ is the total time limit. M is the set of agricultural machines. N is the set of operation points.

## 2.2 Solution framework

To address the complexity and multi-level characteristics of the aforementioned problems, this study proposes a hierarchical, progressive solution framework, as illustrated in Fig 2. This framework combines top-down decomposition with bottom-up integration strategies, decomposing the complex farm preparation operation problem into three interconnected levels of optimization.

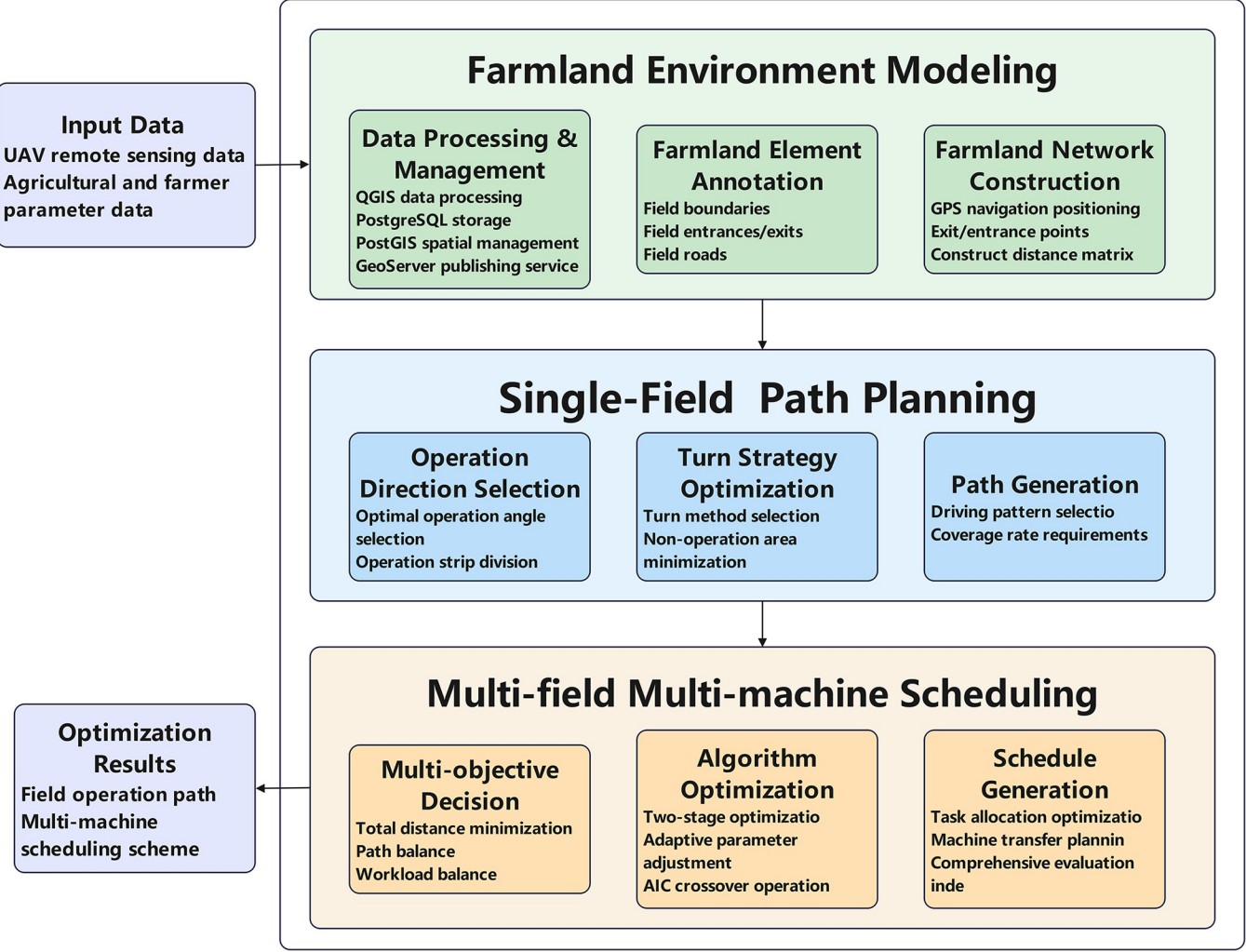

**Fig 2. Schematic diagram of the technical framework for agricultural machinery path planning and task allocation.**

At the environmental modeling layer, high-precision farmland environment models are established through the integration of UAV remote sensing and ground measurements, providing fundamental data support for subsequent optimization. The path planning layer achieves efficient single-field coverage by selecting optimal operational directions and turning strategies, integrating field-specific constraints and operational requirements. At the task allocation layer, the enhanced BNSGA-III algorithm is employed to optimize multi-machine collaborative scheduling, effectively balancing workload distribution and operational efficiency.

The interaction between these layers is facilitated by continuous and integrated data transfer. Environmental models provide data support and spatial constraints for path planning, while path planning results offer workload estimates for task allocation. Scheduling solutions offer feedback to guide parameter optimization, enabling continuous improvement in overall performance. The hierarchical, progressive framework design reduces problem complexity while ensuring optimal performance at the global level.

The detailed methodologies are presented in subsequent sections: Section 2 introduces high-precision modeling of farmland environments; Section 3 elaborates on single-field

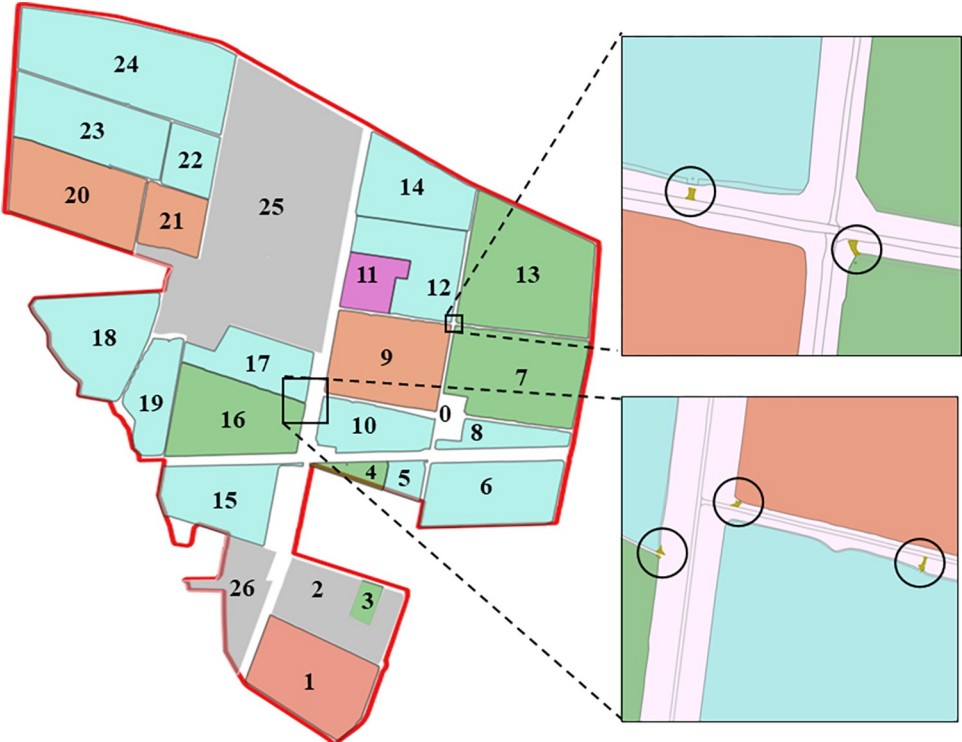

**Fig 3. Huaxing Farm field annotation (circle mark indicates the location of the entrance/exit).**

operation path planning algorithms; and Section 4 focuses on multi-machine collaborative scheduling strategies based on the improved BNSGA-III algorithm.

## 3. Farmland environment modeling

The establishment of an accurate data model for agricultural land is a fundamental step towards achieving comprehensive coverage for path planning and autonomous machinery operations, thereby enhancing overall efficiency. In September 2023, the research team employed a DJI Mavic 3 drone to conduct a survey at Huaxing Farm in Xinjiang, China (44°13′21″N and 87°17′38″E). The field studies were conducted on private agricultural land with explicit permission from the landowners. No specific permits were required as the research involved standard farming practices and did not involve protected species or conservation areas. The aerial survey was conducted using a UAV with the following parameters: flight altitude of 110 m, forward overlap of 80%, side overlap of 70%, and flight speed of 15 m/s. The collected data was subsequently processed into a CSV file and filtered using QGIS for mapping purposes. The refined map data were then stored in a PostgreSQL database and published via GeoServer, enabling access to high-precision maps and related services. Detailed labelling was applied to the data, including the farm boundaries, individual field boundaries, field roads, and field entrances, This is illustrated in Fig 3. The dataset comprises one main warehouse (labelled 0), 24 individual fields, one tourist attraction (labelled 25), and one livestock area (labelled 26). The specific details of the dataset are provided in Table 1.

   Traditional methods for determining distances between farmlands often rely on point-to-point straight-line or great-circle distances. However, such methods fail to account for real-world possibility issues, such as ditches between plots, and do not reflect the actual travel

**Table 1. Partial field data of Huaxing Farm.**

| No. | Code | Field Name | Area (mu) | Boundary Coordinates | Entry/Exit Coordinates |
|---|---|---|---|---|---|
| 1 | w7 | West 7 | 1349.116 | 87.27974618 44.23320304 87.27971784 44.2332726 87.2797184 44.2333898 87.27978011 44.23379365 87.27982333 44.23401292 87.27984274 44.23413331 87.27987075 44.23434025 87.2799457 44.23468577 87.2800137 44.23503366 87.28007132 44.23535154 . . . . . .44.23270913 87.28092859 44.23274463 87.28047582 44.232889 87.28027395 44.23295959 87.28011667 44.23300655 87.27979145 44.23311059 87.27975917 44.23312187 87.2797804 44.23316527 87.27974618 44.23320304 | 87.29799916,44.2063131 87.2960373,44.2011371 87.30319661,44.19651881 87.30637598,44.20233105 |

distance of agricultural machinery. As a result, there can be significant discrepancies between the calculated distances and actual conditions. To address this limitation, this study utilises the coordinate data of plot entrances and exits, and leverages GPS navigation systems to determine the transfer distances between plots for agricultural machinery. For plots with multiple entrances or exits, the one yielding the shortest transfer distance is selected, thereby constructing an accurate distance matrix between plots.

## 4. In-field operation path planning

Efficient in-field operation path planning is essential for identifying cost-effective routes that meet operational requirements while minimizing expenses. In fields with complex and irregular topography, challenges such as insufficient coverage and redundant paths often arise, leading to increased operational time. Therefore, rational path planning is crucial for improving the efficiency of agricultural machinery and reducing the associated operational costs.

### 4.1 Selection of driving pattern

Common full-coverage driving patterns for farmland include back-and-forth, spiral, and headland patterns. The headland pattern is particularly suitable for irregular or complex field shapes. The diverse, irregular boundaries and obstacles of farmland present challenges for traditional patterns, such as back-and-forth and spiral, in achieving efficient coverage [26]. Additionally, tractors have large turning radii, and in irregular fields, the back-and-forth pattern often results in excessive overlaps and uncovered areas due to continuous turns. In contrast, the alternating headland pattern is more adaptable to varying field shapes, reducing the number of turns and overlaps, thereby enhancing operational efficiency. Considering the characteristics of farmland terrain and tractor performance, this study adopts the alternating headland driving pattern.

### 4.2 Selection of operation direction and strip division

The selection of the operational direction significantly impacts the number of turns and strip lengths, consequently affecting energy consumption and operational efficiency. As illustrated in Fig 4, a 180˚ working angle substantially reduces the number of strips and turns compared to a 90˚ angle. To determine the optimal working angle (θ), this study considers factors such as implement width, turning radius, field shape, and objectives of energy conservation and emission reduction.

The first step is to determine the geometric position of each operational path, followed by calculating the total length of all paths. Using the operation angle θ and the initial point of machinery operation $A_1(x_1, y_1)$, the intersection point of each path with the plot boundary, $B_1(x_1', y_1')$, can be determined, thus deriving the linear equation of the tractor's path. The

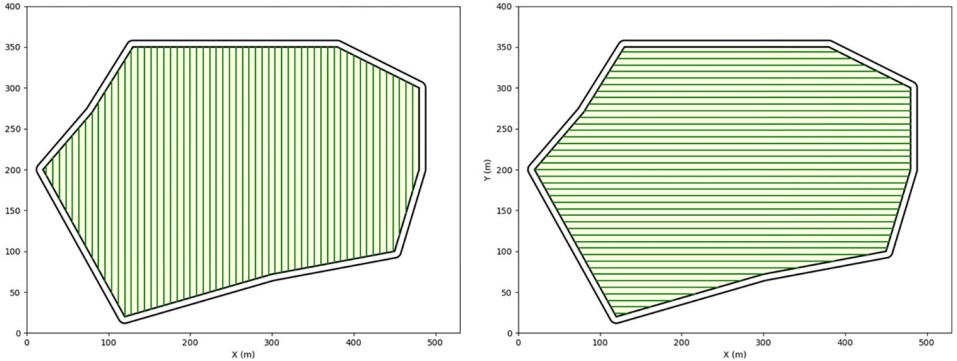

**Fig 4. Different operating angles θ = 90˚, θ = 180˚.**

equation is presented as follows (9).

$$y - y_1 = \tan\theta\left[x - x_1 - \frac{(n-1)\omega}{\sin\theta}\right] \tag{9}$$

By calculating the total length of the tractor's straight-line movement at the intersection points with the field boundary, the overall straight-line distance covered by the tractor can be determined. The total path length is calculated using Eq 10.

$$P_W = \sum_{n=1}^{N+1}\sqrt{(x_n - x'_n)^2 + (y_n - y'_n)^2} \tag{10}$$

Where n represents the path number, and w is the distance between adjacent paths.

## 4.3. Turning methods

The most common turning methods are semi-circular turns, arched turns, pear-shaped turns, and fishtail turns, as shown in Fig 5. **The ABCD represents the sequential trajectory of the turning motion.** When the machine's minimum turning radius equals the width (W), the semicircular turn method is selected. If the turning radius (r) is less than W/2, the arched turn method is used. If r exceeds W/2, either the pear-shaped or fishtail turn method is applied. The

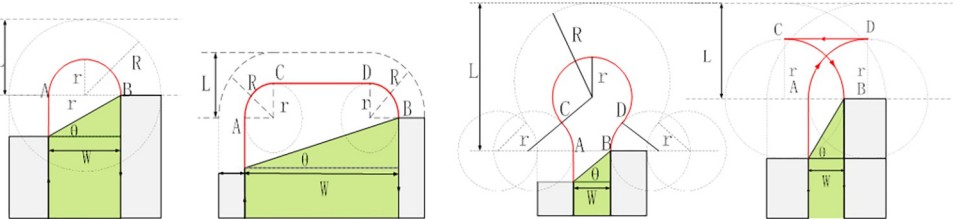

**Fig 5. Common turning methods.**

operational costs ($C_i$) for these turning methods are provided in Eq (11).

$$\begin{cases} C_1 = W\,tan\,\theta + \dfrac{W\pi}{2} \\ C_2 = W\,tan\,\theta + W - 2r + r\pi \\ C_3 = \pi r + W\,tan\,\theta + 4r\,cos^{-1}\left(\dfrac{W+2r}{4r}\right) \\ C_4 = W\,tan\,\theta + (\pi+2)r - W \end{cases} \tag{11}$$

The required headland turning widths, $L_i$, for the various turning methods are given in Eq (12).

$$\begin{cases} L_1 = R\left(1 + cos\,\theta\right) + \dfrac{W}{2} \end{cases}$$

$$L_2 = R(1 - cos\,\theta) + W\left(\dfrac{1}{2} + cos\,\theta\right)$$

$$L_3 = sin\,\theta\sqrt{(2R)^2 - \left(R + \dfrac{\omega}{2}\right)^2} + \dfrac{W\,cos\,\theta}{2} + R + \dfrac{W}{2}$$

$$L_4 = R(sin\,\theta + cos\,\theta) + \dfrac{W}{2} \tag{12}$$

Where W represents the operating row spacing, $\theta$ represents the angle between the operating direction and the field boundary, and R represents the actual turning radius of the agricultural machine. The variable i denotes the turning method, with 1 for semi-circular turns, 2 for arched turns, 3 for pear-shaped turns, and 4 for fishtail turns.

The selection of a turning method can minimize the occurrence of sharp angles, which significantly slow down the machine and increase the time required to complete the task. Additionally, sharp turns expand the actual turning area, leading to overlapping operations and missed spots, ultimately reducing work efficiency. Therefore, a smoother arched turning method, combined with Dubins curve optimization [27], can effectively reduce the number of sharp turns and mitigate the inefficiencies caused by traditional turning methods.

In summary, the total path P for a tractor operating in a polygonal field is $P = P_W + C_i$.

## 5. Multi-machine multi-field collaborative operation scheduling path planning

In the context of multi-machine, multi-field collaborative operation scheduling, it is essential to consider factors beyond inter-field transfer distances, such as the distribution of tractor workloads. Focusing solely on minimizing travel distance may lead to an imbalance in workloads, with some tractors being assigned the majority of tasks while others handle only a few. Conversely, prioritizing load balancing alone may introduce additional travel routes, thereby increasing total energy consumption. Given the considerable variability in task volumes and operating environments across different machines and locations, it is not feasible to predefine the trade-offs between multiple objectives [28].

To address this challenge, a novel solution to the MTSP is proposed, based on enhancements to the NSGA-III algorithm, as shown in Fig 6. The algorithm utilizes Pareto dominance to achieve an optimal balance between conflicting objectives, thereby improving workload

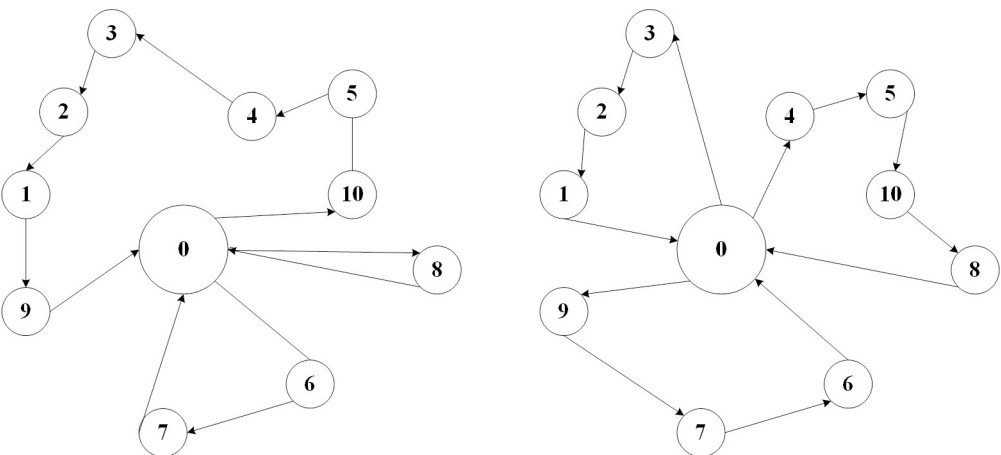

**Fig 6. MTSP route planning diagram standard MTSP, multi-objective optimized MTSP.**

distribution without compromising operational efficiency. This approach enhances the overall operational efficiency and energy utilization of agricultural machinery in real-world applications.

### 5.1. Algorithm description

**5.1.1. NSGA-III algorithm.** As agricultural fields become larger and more mechanized, the decision-making space grows significantly more complex. The NSGA-III algorithm [29] was developed specifically to address high-dimensional, multi-objective optimization problems. Building on the framework of the Nondominated Sorting Genetic Algorithm II (NSGA-II) [30], NSGA-III incorporates a reference point mechanism, which notably reduces the computational burden associated with selecting the Pareto optimal set. However, further enhancements are necessary for its application in agricultural contexts. The NSGA-III algorithm is applied to the problem of multi-machine task allocation in agriculture, with the key stages outlined as follows:

**Initialization Phase**: The structured reference points $\{\lambda^1, \lambda^2, \ldots, \lambda^H\}$ are generated using the Das and Dennis method. An initial population of size N is randomly generated, and the ideal point $z^*$ is initialized. The ideal point is defined as $z^* = (z_1^*, z_2^*, \ldots, z_m^*)$, where $z_i^* = min_{x \in P0} f_i(x)$.

**Evolutionary Operations**: At generation ttt, the offspring population Qt is generated through tournament selection, simulated binary crossover (SBX), and polynomial mutation. The parent population Pt and the offspring population Qt are merged to form the new population Rt, which is then subjected to a non-dominated sorting procedure to obtain the fronts F1, F2, . . .

**Reference Point Association**: Once the ideal point $z^*$ and the nadir point $z^{nad}$ have been updated, the objective functions are normalized. The normalization is performed using the equation:

$$z^n = \frac{f(x) - z^*}{z^{nad} - z^*} \tag{13}$$

The perpendicular distance between individual x and reference point λj is then calculated as:

$$d_\perp(x, \lambda^j) = \|z^n - (\lambda^j \cdot z^n)\lambda^j\|) \tag{14}$$

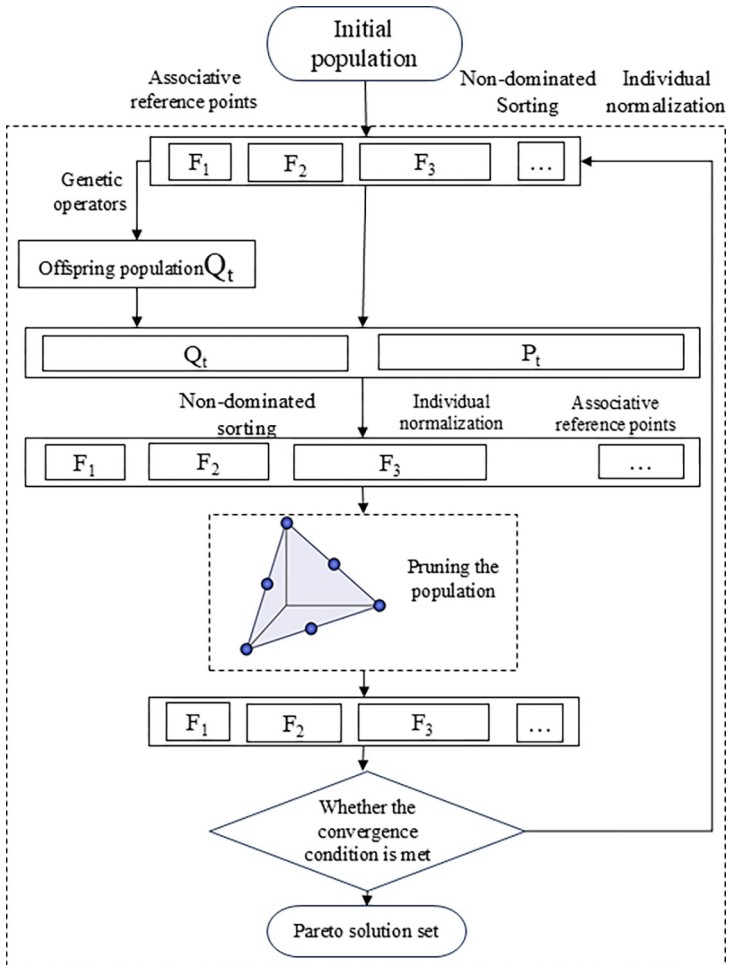

**Fig 7. NSGA-III algorithm flow chart.**

Each individual is subsequently associated with the reference point that exhibits the minimum perpendicular distance.

**New Population Selection:** Individuals are selected based on their non-dominated ranks until the front Fl is no longer entirely included. For individuals within Fl, the niche count $\rho j$ is calculated for each reference point. The selection process prioritizes reference points with the lowest $\rho j$, and individuals with the smallest $d\perp$ are chosen from those associated with the selected reference point. This process continues with updated $\rho j$ values until the population size reaches N.

The algorithm maintains solution diversity in high-dimensional objective spaces through the reference point mechanism while ensuring convergence through non-dominated sorting. The introduction of crossover and mutation operations enhances population diversity and prevents premature convergence. During each iteration, the algorithm prioritizes individuals near reference points with lower niche counts, ensuring a balanced coverage of objectives in the task allocation process. The algorithm terminates when the predefined stopping criteria are met, outputting the final Pareto optimal solution set. The complete NSGA-III process is illustrated in Fig 7.

**5.1.2 BWO algorithm.** The method integrates the Beluga Whale Optimization (BWO) algorithm [31] with NSGA-III for initial population generation and subsequent fine-tuning during the optimization phases. The BWO algorithm emulates the hunting behavior of beluga whales, encompassing exploration, exploitation, and whale fall phases. During the exploration phase, large-scale searches are conducted through the synchronized and mirrored behaviors observed in beluga whales. The exploitation phase utilizes a Levy flight to enhance convergence, while the whale fall phase achieves fine-tuning through dynamic step size adjustment. This multi-stage strategy is particularly well-suited to the hierarchical optimization requirements of farmland path planning. Compared to traditional genetic algorithms [32] and Pigeon-Inspired Optimization [33], BWO offers several advantages, including fewer parameters, adaptive adjustment mechanisms, and superior capability to escape local optima. This enables a better balance between global exploration and local exploitation. The process flow is illustrated in Fig 8.

Where the algorithm incorporates two adaptive variables: the balance factor $B_f$ and the whale fall probability $W_f$. $B_f$ enables a smooth transition between the exploration and exploitation phases, effectively balancing the global and local search processes. The mathematical model is represented by Eq (16):

$$\begin{cases} B_f = B_0(1 - {}^t/_{2T}) \\ W_f = 0.1 - 0.05^t/_T \end{cases} \tag{15}$$

Where t represents the current iteration, T is the maximum number of iterations, and $B_0$ is a random number between 0 and 1. When $B_f > 0.5$, the algorithm enters the exploration phase; when $B_f \leq 0.5$, it enters the exploitation phase; and when $B_f < W_f$, it enters transitions to the whale fall phase.

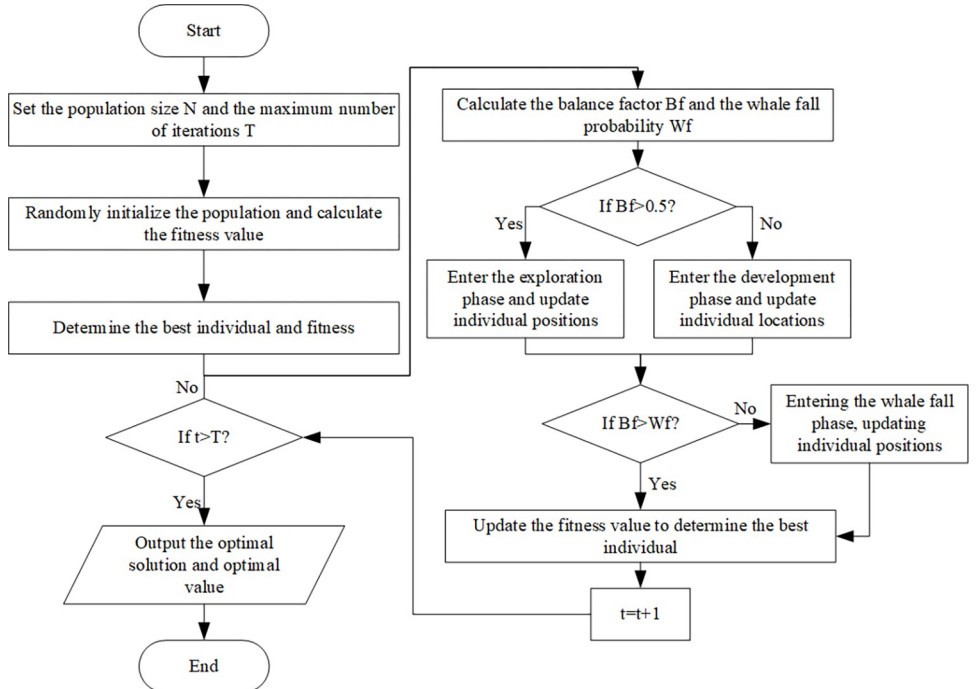

**Fig 8. Beluga Whale Optimization algorithm flow chart.**

(1) Exploration Phase:

$$
\begin{cases}
X_{i,j}^{t+1} = X_{i,pj}^t + (X_{r,p1}^t - X_{i,pj}^t)(1 + r_1)\sin(2\pi r_2) \quad j = even \\
X_{i,j}^{t+1} = X_{i,pj}^t + (X_{r,p1}^t - X_{i,pj}^t)(1 + r_1)\cos(2\pi r_2) \quad j = odd
\end{cases}
\tag{16}
$$

The variable $X_{i,j}^{t+1}$ represents the updated position of the current i-th beluga whale in the j-th dimension, while $p_j$ is a randomly selected dimension from the set of dimensions (j = 1,2.,d); $X_{i,pj}^t$ and $X_{r,p1}^t$ represent the current positions of the i-th and r-th beluga whales respectively, $r_1$ and $r_2$ are random numbers between 0 and 1; The trigonometric functions $\sin(2\pi r_2)$ and cos $(2\pi r_2)$ are used to simulate the synchronization and mirror behaviors of beluga whales.

(2) Exploitation Phase:

$$
\begin{cases}
X_i^{t+1} = r_3 X_{best}^t + r_4 + C_1 \cdot L_F (X_r^t - X_i^t) \\
\text{Where :} \\
C_1 = 2r_4 \left(1 - \dfrac{t}{T}\right) \\
L_F = 0.05 \times \dfrac{u \cdot \delta}{|v|^{\frac{1}{\beta}}}
\end{cases}
\tag{17}
$$

Here, $C_1$ represents the random jump intensity measuring Levy flight, $L_F$ is the Levy flight function. $X_r^t$ and $X_i^t$ represent a randomly selected beluga whale and the current position of the i-th beluga whale respectively; $X_i^{t+1}$ represents the updated position of the i-th beluga whale; $X_{best}^t$ represents the best position of beluga whales; $r_3$ and $r_4$ are random numbers between 0 and 1.

(3) Whale Fall Phase:

$$
\begin{cases}
X_i^{T+1} = r_5 X_i^T + r_6 X_r^T + r_7 X_{step}^T \\
\text{Where :} \\
X_{step}^T = (u_a - u_b)\exp\left(-C_2 \dfrac{t}{T}\right) \\
C_2 = 2W_f \cdot n
\end{cases}
\tag{18}
$$

$r_5$, $r_6$, and $r_7$ are random numbers between 0 and 1; $X_{step}^T$ is the step length of whale fall, $C_2$ is the step length factor.

## 5.2 Improvements to the BNSGA-III algorithm

To further enhance NSGA-III's effectiveness in solving multi-objective MTSP, this paper proposes a hybrid optimization algorithm called BNSGA-III. This algorithm combines multiple optimization strategies, fully considering the trade-offs between various objectives during initial solution generation, global search, and local development processes, thereby improving overall optimization performance. Specific improvements include:

**5.2.1. Generation of diverse initial populations.** In the context of multi-objective optimization, the quality of the initial population is of paramount importance, as it significantly influences the overall performance of the algorithm. To enhance the quality and diversity of the initial population, this paper proposes a method that combines BWO with random initialization for the generation of the initial population. BWO emulates the exploration and exploitation behavior of beluga whales during predation, generating initial paths with optimal distribution characteristics for each agricultural machine, thereby accelerating the algorithm's

convergence speed. Simultaneously, a portion of the individuals is generated through random initialization to enhance population diversity and prevent the algorithm from falling into local optima during the early evolutionary stages. The high-quality individuals generated by BWO, combined with the diversity introduced through random initialization, provide the initial population with extensive and balanced coverage of the search space, thereby offering a robust and efficient foundation for subsequent optimization stages. This approach significantly improves the overall effectiveness and convergence efficiency of the algorithm.

**5.2.2. Adaptive inversion crossover.** To address the limitations of traditional crossover operations in maintaining population diversity and exploring new solution spaces, this study proposes the Adaptive Inversion Crossover (AIC) method, as shown in Fig 9. This method introduces a dynamic adjustment mechanism that adaptively adjusts crossover operations according to the needs of different iteration stages during the optimization process, thereby balancing global exploration and local exploitation, improving solution quality and convergence speed in solving MTSP. The specific steps of AIC are as follows:

1. Dynamically calculate the subsegment length $\lambda(t)$.

2. Randomly select a subsegment from the parent chromosome with a length of $L*\lambda(t)$, where L is the total length of the chromosome.

3. Decide whether to perform inversion operation on the selected subsegment based on the inversion probability $Pr(t)$.

4. Insert the processed subsegment into a random position of another parent chromosome to generate new offspring.

**1.Chromosome Encoding**

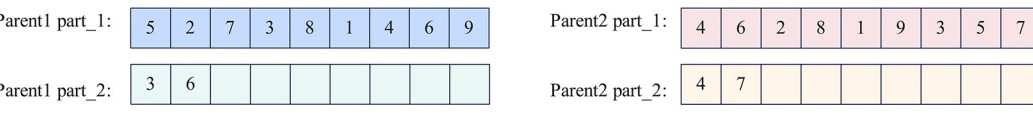

**2.RSRR Crossover**

Step1:Select segment from Rarent 1(length based on $\lambda(t)$)

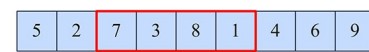

Step2:Conditionally reverse segment (based on Pr(t))

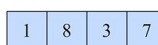

Step4:Insert reverse segment into Parent2

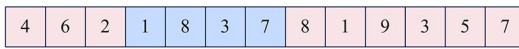

**3. Repair Process**

Remove duplicates and adjust

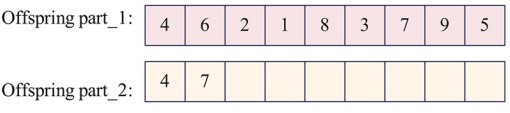

**4.GraphicalRepresentation of Offspring Routr**

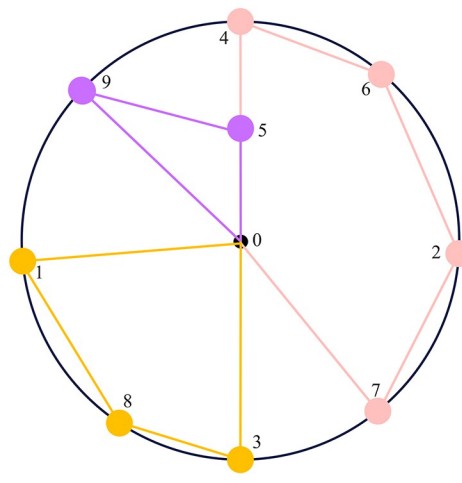

**Fig 9. Schematic diagram of AIC operation.**

5. Repair the offspring to ensure the validity of the generated solution in the MTSP.

In this process, the dynamic subsegment length factor λ(t) is used to dynamically adjust the length of the subsegment selected in the crossover operation. In the early stages of optimization, longer subsegments help with extensive global exploration; in the later stages, as iterations progress, the subsegment length gradually decreases, allowing for more refined local optimization. The formula is as follows:

$$\lambda(t) = \lambda_{max} - (\lambda_{max} - \lambda_{min})\left({}^{t}/{}_{T}\right)^{\alpha} \tag{19}$$

Where $\lambda_{max}$ and $\lambda_{min}$ are the maximum and minimum proportions of subsegment length, respectively; T is the total number of iterations; α is a parameter controlling the decay rate; t is the current iteration number.

The adaptive inversion probability $P_r(t)$ controls the frequency of inversion operations. Since more global exploration is needed in the early stages of optimization, longer subsegments are selected, so the probability of applying inversion operations is lower; while in the later stages of optimization, as the subsegment length shortens, the inversion probability gradually increases, thus more frequently inverting subsegments to enhance local search capabilities. The formula for this mechanism is:

$$P_r(t) = 1 - \lambda(t) \tag{20}$$

AIC effectively achieves dynamic balance in optimization by flexibly adjusting crossover behavior during the optimization process. In the early stages of optimization, it enhances global exploration capabilities of the solution space through longer subsegments, while in later stages, shorter subsegments support more refined local searches. As iterations progress, the inversion probability is also dynamically adjusted, ensuring detailed development of solutions in the later stages of optimization, thus balancing exploration and exploitation. Through these methods, AIC can effectively maintain population diversity at various stages of optimization, avoiding premature convergence of the algorithm, particularly suitable for solving complex MTSP, significantly improving the quality of solutions in the path optimization process.

**5.2.3. Two-stage optimization strategy of BNSGA-III.** After generating the initial population, BNSGA-III adopts a two-stage optimization strategy, including the NSGA-III global optimization stage and the BWO fine-tuning stage, jointly enhancing the algorithm's optimization performance.

(1) NSGA-III global optimization stage

NSGA-III achieves a good trade-off between multiple objectives through non-dominated sorting and reference point-based selection mechanisms. To address the local optimum trap problem common in farmland path planning, this paper introduces adaptive reference point generation and local search strategies.

The adaptive reference point generation method dynamically adjusts the number of reference point divisions in each iteration based on the generation number, reflecting the transfer situation of agricultural machines between different fields, avoiding premature population concentration, maintaining diversity, and improving the convergence speed and quality of solutions. Moreover, periodically regenerating reference points ensures their consistency with population distribution, enhancing the algorithm's flexibility and optimization performance. In this process, the number of reference point divisions Kt is dynamically adjusted with

generation t, as shown in the following equation:

$$K_t = K + \left\lfloor \frac{t}{T/10} \right\rfloor \tag{21}$$

Where K is the initial number of reference point divisions, t is the current generation number, and T is the maximum number of iterations. BNSGA-III achieves a balance between global search and local search at different evolutionary stages by dynamically adjusting crossover probability Pc and mutation probability Pm, thereby improving the algorithm's convergence speed and solution quality. In the early stages of evolution, to explore more possible path combinations, higher crossover probability and lower mutation probability are set to promote solution diversity; in the later optimization stages, crossover probability is gradually reduced and mutation probability is increased to accelerate convergence speed and enhance local development capabilities, reducing the generation of ineffective paths.

$$\begin{cases} P_c = P_{c0} - (P_{c0} - P_{cmin}) \cdot \dfrac{t}{T} \\ P_m = P_{m0} - (P_{mmax} - P_{m0}) \cdot \dfrac{t}{T} \end{cases} \tag{22}$$

Where $P_c$ is the crossover probability, $P_m$ is the mutation probability, t is the current iteration number, T is the total number of iterations, $P_{c0}$ and $P_{m0}$ are the initial crossover and mutation probabilities, and $P_{cmin}$ and $P_{mmax}$ are the minimum crossover probability and maximum mutation probability, respectively.

(2) BWO fine-tuning optimization stage

After NSGA-III global optimization, BNSGA-III introduces the BWO fine-tuning optimization stage to further improve solution quality. The BWO fine-tuning stage is mainly used for in-depth development of current solutions, focusing on optimizing individuals close to the Pareto front. By simulating the hunting behavior of whales, BWO dynamically adjusts the positions of individuals, making them continuously approach the optimal reference points, thereby improving the accuracy and balance of solutions. This stage adaptively adjusts the falling probability of whales and the Lévy flight step length, combined with local development operations, achieving large-scale exploration and fine-tuning in the search space. This mechanism effectively maintains population diversity while accelerating the convergence of solutions towards the Pareto optimal front. Through the two-stage optimization strategy, BNSGA-III conducts global exploration through NSGA-III in the initial stage, establishing a good population distribution; in the later stage, it performs fine-tuning optimization through BWO, improving the local optimality of solutions. The combination of adaptive reference point generation and adaptive parameter adjustment enables the algorithm to achieve a good balance between global exploration and local development, effectively improving solution quality and convergence speed. The pseudo-code for the BNSGA-III algorithm is as follows.

```
Algorithm: BNSGA-III
Input: N: population size
  T: maximum iterations
  M: number of objectives
  div: initial division parameter
Output: PS: Pareto optimal set
/* Phase 1: Initialization */
1: P0 ← BWO_Initialize(⌊0.8N⌋) ∪ Random_Initialize(N - ⌊0.8N⌋)
2: evaluate(P0)
3: t ← 0
```

```
/* Phase 2: NSGA-III Global Search */
4: while t < T/2 do
5:  /* Generate reference points */
6:  Z, base_Z ← adaptive_reference_points(Pt, M, div, t)
7:  F ← non_dominated_sort(Pt)
8:  Pt ← niching_selection(F, N, base_Z)
9:  /* BWO local search */
10: for i = 1 to min(3, |F|) do
11:   for each 5th individual in Fi do
12:     Fi[j] ← bwo_local_search(Fi[j], Z)
13:   end for
14: end for
15: /* Offspring generation */
16: pc ← pc_max−(pc_max−pc_min)·t/(T/2)
17: pm ← pm_min + (pm_max−pm_min)·(t/(T/2))²
18: Qt ← ∅
19: while |Qt| < N do
20:   parents ← tournament_select(Pt)
21:   if random(0,1) < pc then
22:     /* Adaptive Island Crossover (AIC) */
23:     λt ← λmax−(λmax - λmin)·(t/T)^α
24:     Pr ← 1 - λt /* Reverse probability */
25:     if random(0,1) < 0.5 then
26:       offspring ← AIC_crossover(parents, λt, Pr)
27:     else
28:       offspring ← traditional_crossover(parents)
29:     end if
30:     offspring ← mutate(offspring, pm)
31:     Qt ← Qt ∪ {offspring}
32:   end if
33:   end while
34: Rt ← Pt ∪ Qt
35: F ← non_dominated_sort(Rt)
36: Pt+1 ← niching_selection(F, N)
37: t ← t + 1
38: end while
/* Phase 3: BWO Fine-tuning */
39: while t < T do
40: Z, base_Z ← adaptive_reference_points(Pt, M, div, t)
41: P' ← BWO_guided_search(Pt, base_Z) /* Enhanced exploitation */
42: F ← non_dominated_sort(P')
43: Pt+1 ← niching_selection(F, N)
44: t ← t + 1
45: end while
46: return get_pareto_front(PT)
```

## 6. Simulation analysis

To validate the performance of the BNSGA-III algorithm in the context of agricultural multi-robot task allocation, a series of simulation experiments were conducted in a controlled equipment environment. The experimental setup comprised an Intel(R) Core(TM) i7-13700H processor with 16GB of RAM, running on a Windows 10 operating system, with Python as the development language. Algorithm parameter settings: population size: 100, maximum number of iterations: 100, crossover probability: $Pc = 0.9$, mutation probability: $Pm = 0.1$, BWO algorithm parameters: $B0 = 0.5$, $\alpha = 0.6$, AIC operating parameters: $\lambda max = 0.8$, $\lambda min = 0.2$. The

**Table 2. Planning data for different plots.**

| Plot Number | Random Direction | | | Short Path | | | Fewer Turns | | |
|---|---|---|---|---|---|---|---|---|---|
| | Effective Path Length | Number of Turns | Coverage Rate | Effective Path Length | Number of Turns | Coverage Rate | Effective Path Length | Number of Turns | Coverage Rate |
| Plot 1 | 23620.8 | 41 | 94.3% | 23166.2 | 33 | 94.3% | 23166.2 | 33 | 95.7% |
| Plot 2 | 24003.3 | 48 | 95.0% | 23701.2 | 42 | 95.0% | 23701.2 | 39 | 96.1% |
| Plot 3 | 24032.7 | 47 | 96.0% | 23597.7 | 42 | 95.0% | 23864.3 | 42 | 96.1% |
| Plot 4 | 23872.5 | 50 | 95.8% | 23578.4 | 38 | 95.0% | 23578.4 | 38 | 96.0% |

experimental data were derived from actual field coordinates and operational data from Huaxing Farm.

## 6.1. Single field experiment

The objective of the single-field plot experiments was to evaluate the effectiveness of different path planning strategies in reducing the number of turns and improving the coverage rate. Four distinct plots were employed in the experiments, with the optimal operational direction determined through an exhaustive search algorithm. The experimental results are presented in Table 2. Compared to random operational directions, this method demonstrated a reduction in path length of 1.9% to 3.1%, a decrease in the number of turns of 19.5% to 24.0%, and an improvement in the coverage rate of 1.0% to 1.4%. The experiments revealed that the optimal path planning method effectively reduced path length and the number of turns across all test plots, thereby enhancing the coverage rate. The results for the Huaxing Farm plot are shown in Fig 10.

## 6.2. Collaborative scheduling of multiple agricultural machines in multiple fields

The objective of this experiment is to evaluate the performance of different algorithms when multiple tractors work simultaneously and to assess the effects of different path planning schemes in terms of minimizing the total driving distance, balancing the driving distance between tractors, and balancing the workload. The experiment focuses on scheduling collaborative operations involving multiple agricultural machines and multiple fields. The effectiveness of the enhanced algorithms is validated through a comparative and analytical evaluation of four algorithms: BNSGA-III, NSGA-II, NSGA-III, and MOEA/D [34]. The experiment tests scenarios involving three, four, five, and six farm machines, respectively. The evaluation metrics include total distance, path balance, workload balance, HV, IGD, and Spread values. Each

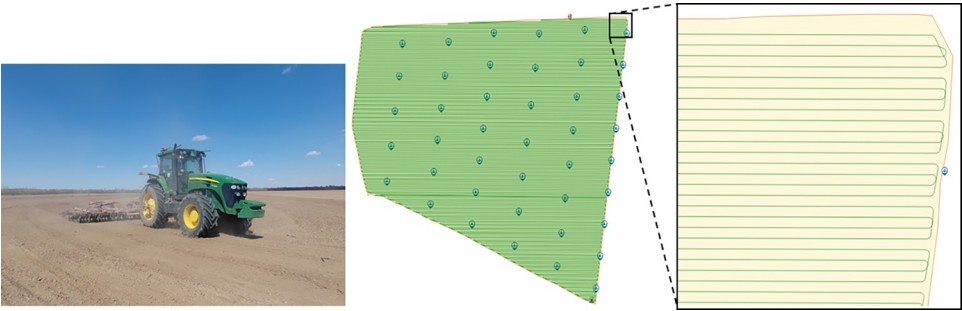

**Fig 10. Agricultural field path planning diagram.**

**Table 3. Algorithm comparison.**

| Algorithm | Machines (units) | Scheduling Path | Distance (km) | Workload (mu) | Total Distance (km) | Distance Balance (km) | Workload Balance (mu) |
|---|---|---|---|---|---|---|---|
| BNSGA-III | 1 | 0 -> 18 -> 15 -> 3 -> 16 -> 8 -> 10 -> 12 -> 11 -> 7 -> 0 | 19.04 | 4612.45 | 59.46 | 4.61 | 5.26 |
| | 2 | 0 -> 19 -> 6 -> 20 -> 21 -> 17 -> 1 -> 13 -> 0 | 25.82 | 4607.19 | | | |
| | 3 | 0 -> 14 -> 24 -> 23 -> 22 -> 2 -> 5 -> 4 -> 9 -> 0 | 14.6 | 4608.02 | | | |
| NSGA-II | 1 | 0 -> 11 -> 22 -> 20 -> 24 -> 23 -> 1 -> 0 | 17.15 | 4591.01 | 74.56 | 13.9 | 53.42 |
| | 2 | 0 -> 3 -> 12 -> 16 -> 19 -> 10 -> 6 -> 14 -> 8 -> 13 -> 0 | 26.37 | 4592.21 | | | |
| | 3 | 0 -> 18 -> 21 -> 15 -> 7 -> 17 -> 5 -> 9 -> 4 -> 2 -> 0 | 31.05 | 4644.44 | | | |
| NSGA-III | 1 | 0 -> 21 -> 19 -> 23 -> 24 -> 11 -> 15 -> 1 -> 5 -> 12 -> 0 | 29.84 | 4615.38 | 69.24 | 11.78 | 24.59 |
| | 2 | 0 -> 9 -> 17 -> 3 -> 18 -> 20 -> 7 -> 16 -> 0 | 21.34 | 4593.84 | | | |
| | 3 | 0 -> 8 -> 10 -> 13 -> 22 -> 14 -> 2 -> 4 -> 6 -> 0 | 18.06 | 4618.44 | | | |
| MOEA/D | 1 | 0 -> 23 -> 10 -> 21 -> 2 -> 1 -> 11 -> 4 -> 24 -> 5 -> 0 | 37.54 | 4501.46 | 90.69 | 13.71 | 70.07 |
| | 2 | 0 -> 9 -> 20 -> 17 -> 12 -> 20 -> 19 -> 23 -> 6 -> 0 | 29.32 | 4433.07 | | | |
| | 3 | 0 -> 21 -> 21 -> 17 -> 15 -> 9 -> 6 -> 24 -> 0 | 23.83 | 4503.15 | | | |

experimental setup was run independently 100 times, and the average value of the evaluation metrics was recorded every 10 iterations to minimize the impact of random fluctuations. The results of the experiment are shown in Table 3 and Figs 11–14.

The experimental results presented in Table 3 demonstrate that the BNSGA-III algorithm outperforms others in terms of total travel distance, distance balance, and workload balance. Compared to NSGA-II, NSGA-III, and MOEA/D, BNSGA-III achieves a reduction in total travel distance of 20.3%, 14.1%, and 34.4%, respectively. Similarly, it reduces transfer distance by 66.2%, 60.9%, and 66.2%, respectively. Furthermore, workload imbalance is reduced by 90.2%, 78.7%, and 92.9%, respectively. These improvements indicate that the enhanced algorithm effectively optimizes the planning of agricultural machinery travel routes, leading to reduced overall transfer distances, operating times, and fuel consumption. Additionally, the improved workload distribution is more balanced, avoiding the overuse of specific machines due to excessive tasks.

The fitness iteration curves in Fig 11 demonstrate that BNSGA-III has superior global search capability and local convergence precision. Furthermore, it converges more rapidly, identifying high-quality solution sets in fewer iterations, with enhanced convergence performance compared to NSGA-II, NSGA-III, and MOEA/D. As the number of agricultural machines increases from three to six, the number of iterations required to reach convergence shows a slight increase. However, BNSGA-III is much less affected than other algorithms, demonstrating superior stability. The final fitness values of BNSGA-III are consistently superior, with improvement rates ranging from 12.3% to 18.7% compared to the second-best algorithm.

Figs 12–14 show that BNSGA-III offers significant advantages across three key metrics: Hypervolume (HV), Inverted Generational Distance (IGD), and Spread. The HV curve

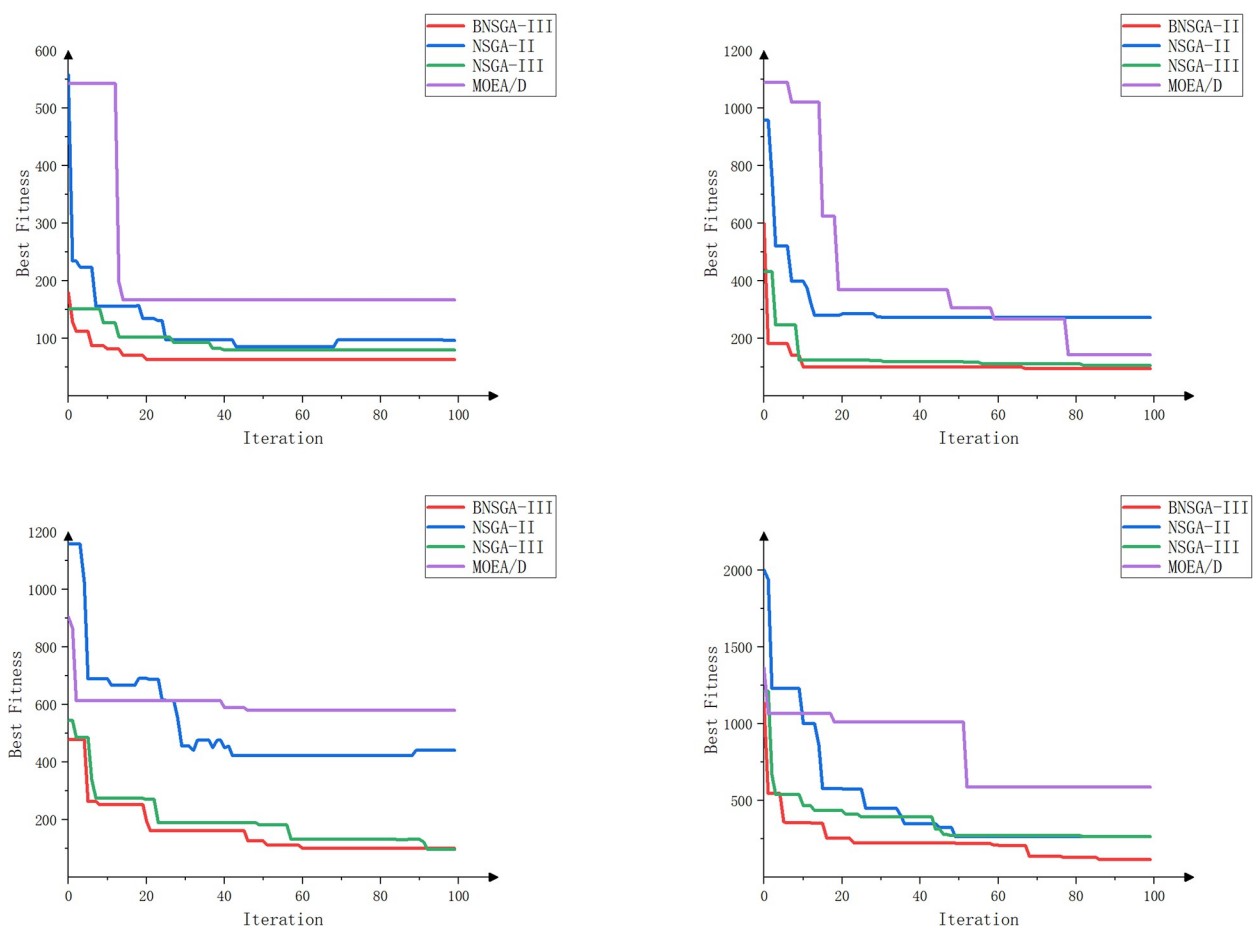

**Fig 11. Iteration curves for agricultural machines with different machine numbers (3, 4, 5, and 6 machines).**

shows that the solution set generated by BNSGA-III covers a larger area, indicating a better ability to identify a more comprehensive and balanced range of solutions on the Pareto front. Compared to NSGA-III, the average HV improved by 15.3%. The IGD curve reflects the average distance between the solution set and the true Pareto front and shows that BNSGA-III consistently achieves lower IGD values, indicating a closer proximity to the optimal solution set. The Spread curve illustrates the distribution of the solution set, with BNSGA-III maintaining a more uniform distribution of solutions across different numbers of agricultural machines, thereby maintaining diversity and avoiding excessive concentration in localized areas.

Overall, BNSGA-III demonstrates superior solution quality and optimization performance across varying numbers of agricultural machines. This indicates its effectiveness in solving multi-objective optimization problems, excelling not only in identifying optimal solutions but also in ensuring the diversity and coverage of the solution set.

## 7. Conclusion

(1) In single-field operations, by analyzing the impact of different operation directions on farmland operation paths and energy consumption, and using brute force search to determine the optimal operation angle, combined with interval row-following and Dubins

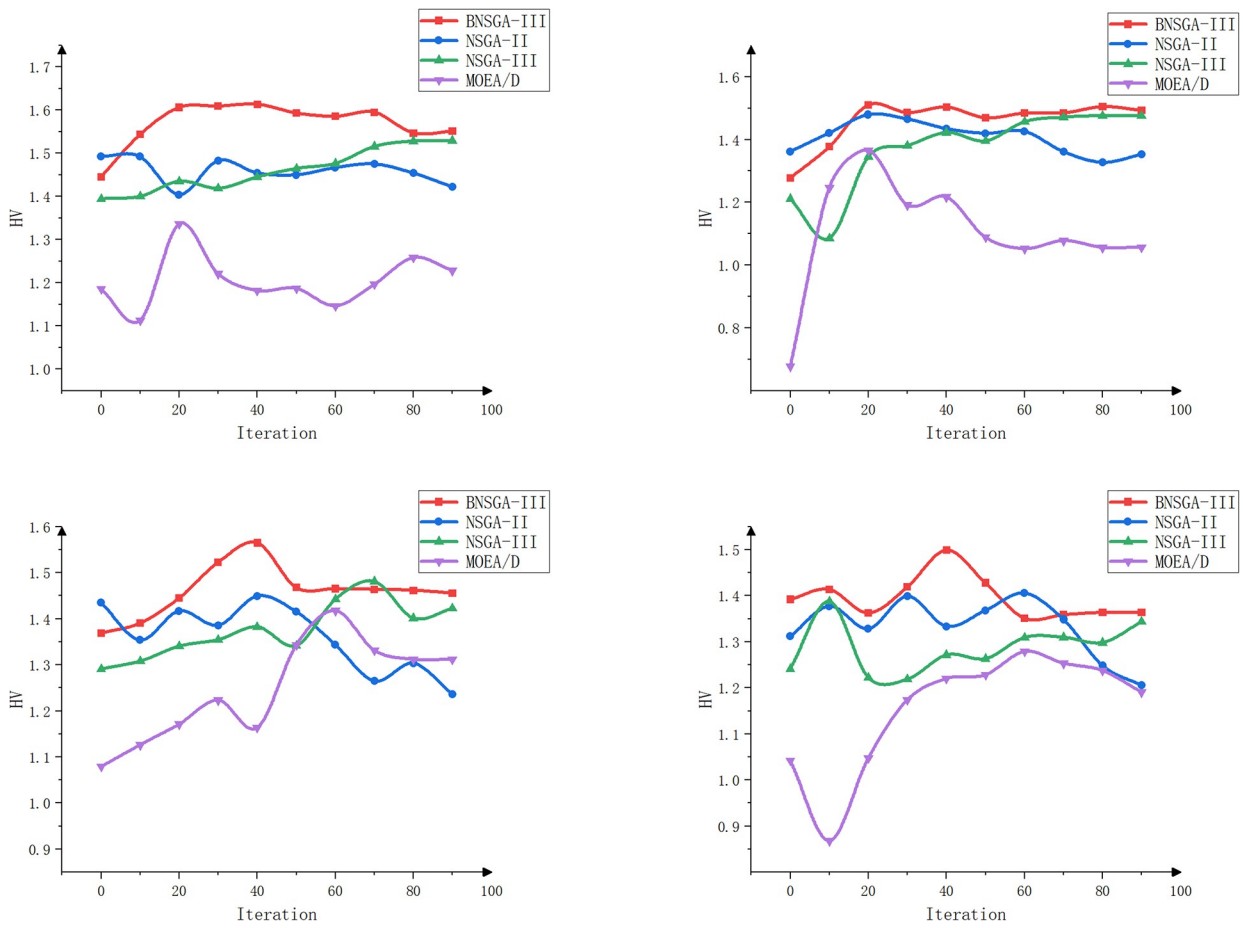

**Fig 12. HV curves for agricultural machines with different machine numbers (3, 4, 5, and 6 machines).**

curve-optimized turning strategies, the field coverage rate was effectively improved, reducing operational energy consumption and time costs. Experimental results show that compared to random operation directions, this method can effectively reduce path length by 1.9%-3.1%, decrease turning frequency by 19.5%-24.0%, and improve coverage by 1.0%-1.4%.

(2) In order to address the issue of multi-machine, multi-field collaborative operation scheduling, this study develops the BNSGA-III algorithm, which effectively balances three distinct optimization objectives: total travel distance, path equilibrium, and workload distribution. The algorithm employs a two-stage optimization strategy. Initially, it improves the quality of the initial population by combining BWO with random initialization. Subsequently, it uses NSGA-III for global exploration, followed by BWO for fine-tuning optimization. To enhance adaptability and search efficiency, an adaptive reference point generation mechanism and an AIC crossover operation are introduced, effectively addressing the path planning and task allocation challenges in the MTSP. The experimental results show that, compared to NSGA-II, NSGA-III, and MOEA/D algorithms, BNSGA-III is able to rapidly identify the Pareto-optimal solution set. This is demonstrated by reductions in total distance of 20.3%, 14.1%, and 34.4%; reductions in transportation distance of 66.2%, 60.9%, and 66.2%; and improvements in workload balance of 90.2%, 78.7%, and 92.9%, respectively. Furthermore,

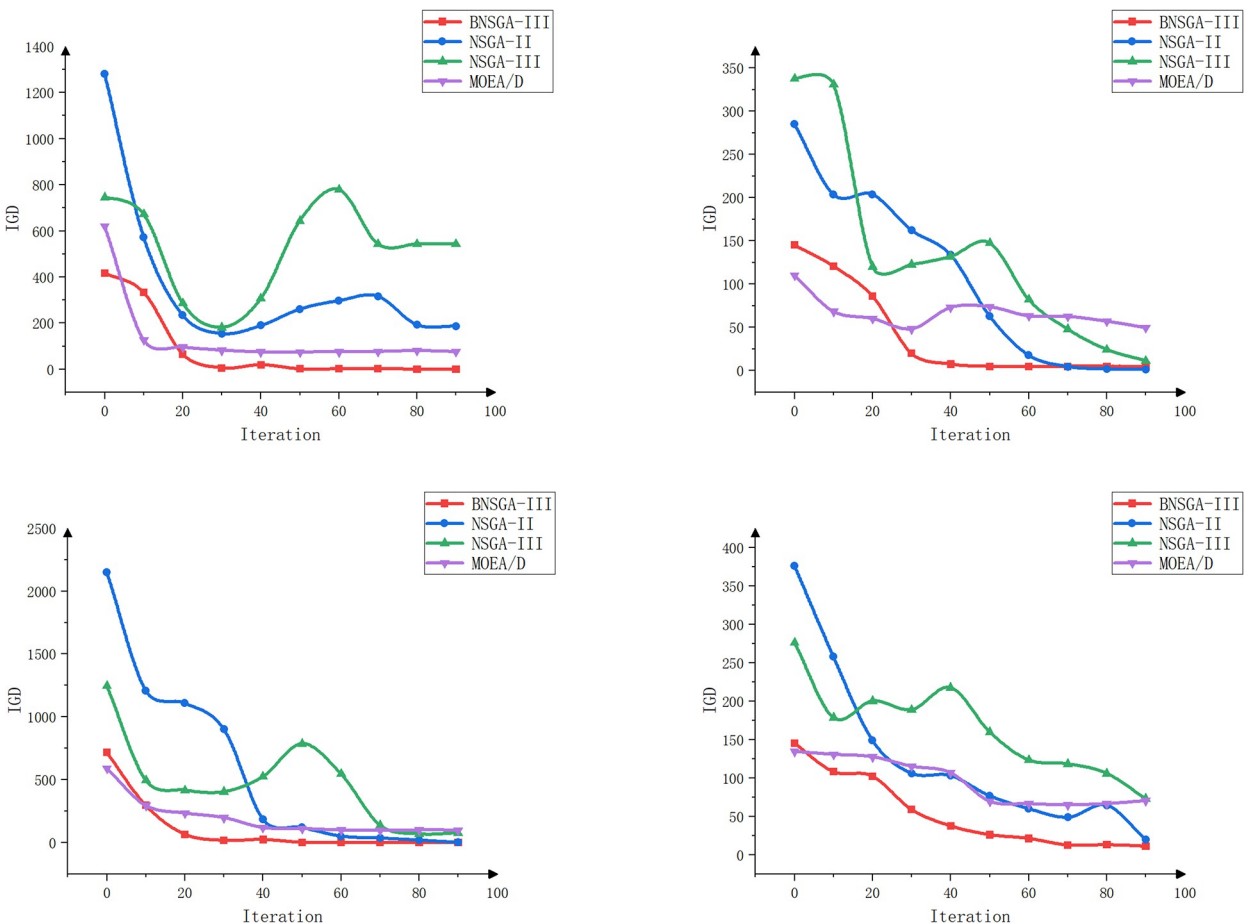

**Fig 13. IGD curves for agricultural machines with different machine numbers (3, 4, 5, and 6 machines).**

BNSGA-III showed improvements of 15.3%, 22.7%, and 18.9% in comparison to NSGA-III on three key performance metrics: HV, IGD, and Spread. These results highlight the algorithm's comprehensive advantages in terms of solution quality, convergence, and diversity. The BNSGA-III algorithm has the potential to significantly enhance agricultural machinery scheduling efficiency and improve the operational effectiveness of unmanned farms in practical applications.

(3) The enhanced algorithm presented in this study effectively reduces fuel consumption, decreases operation time, and improves resource utilization by optimizing agricultural machinery paths and task allocation. This provides farm managers with a valuable decision-support tool, contributing significantly to the development of precision and intelligent agriculture, as well as promoting energy conservation, emission reduction, and sustainable agricultural production. Future research will aim to further enhance the algorithm's performance, expand its application to tasks such as sowing and fertilizing, and support the advancement of intelligent agriculture, enabling comprehensive scheduling and management of unmanned farms.

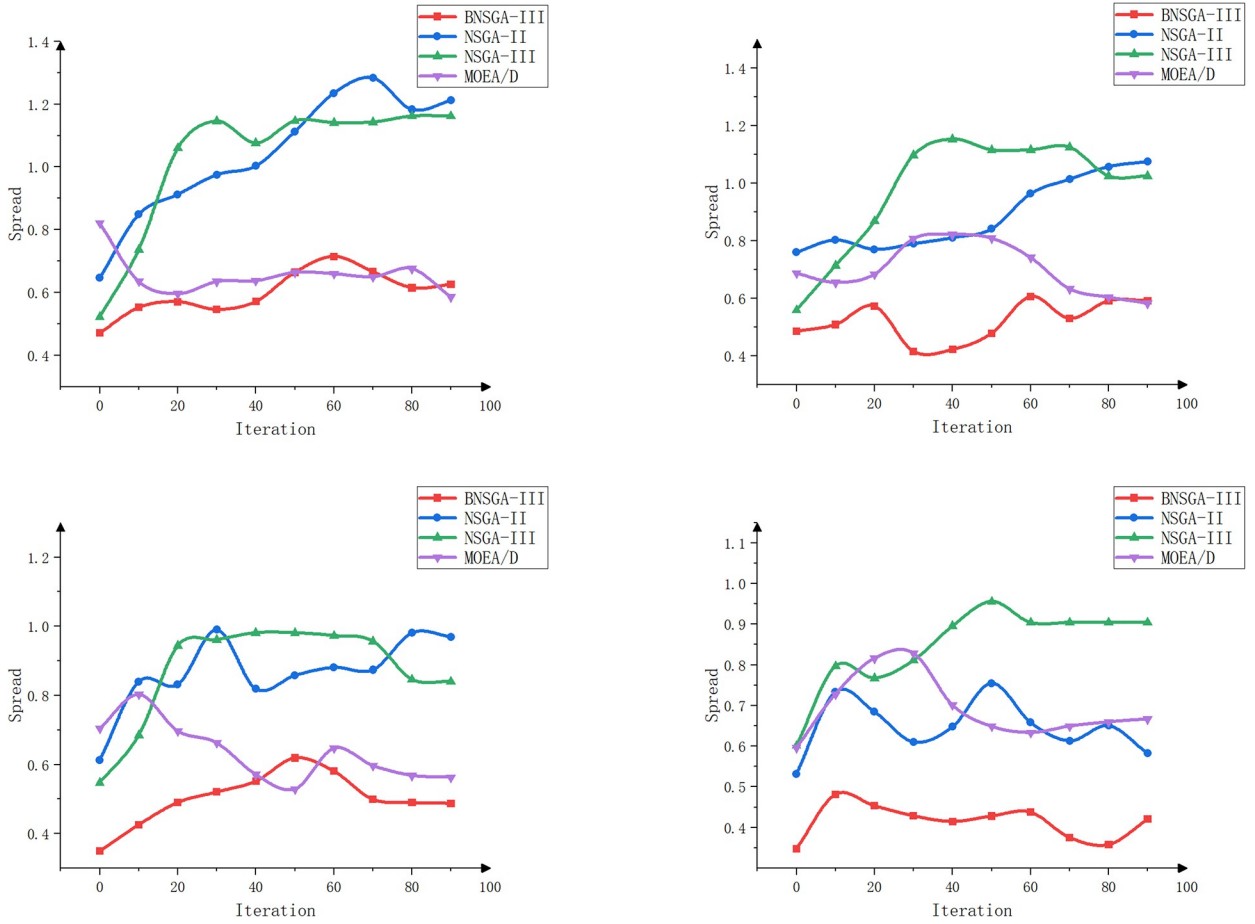

**Fig 14. Spread curves for agricultural machines with different machine numbers (3, 4, 5, and 6 machines).**

## Acknowledgments

We thank all contributors to the study, the project team members for their help with data collection and dataset construction, and the staff at Huaxing Farm for providing access and permission to sample the farmland for data collection.

## Author Contributions

**Conceptualization:** Manxian Yang, Taihong Zhang.

**Data curation:** Manxian Yang, Taihong Zhang.

**Formal analysis:** Yanhong Chen, Yongke Li, Taihong Zhang.

**Funding acquisition:** Yanhong Chen, Yongke Li, Taihong Zhang.

**Investigation:** Manxian Yang.

**Methodology:** Manxian Yang.

**Project administration:** Taihong Zhang.

**Resources:** Yanhong Chen, Yongke Li.

**Software:** Manxian Yang, Tianlun Wu.

**Supervision:** Yanhong Chen, Yongke Li, Taihong Zhang.

**Validation:** Manxian Yang, Yanhong Chen, Yongke Li, Tianlun Wu.

**Writing – original draft:** Manxian Yang.

**Writing – review & editing:** Yanhong Chen, Tianlun Wu.

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
