## [Decision Letter · Decision Letter 0]

7 Nov 2024

PONE-D-24-46846Multi-objective Optimization Method for Path Planning and Task Allocation in Farmland Land Levelling Based on an Improved NSGA-IIIPLOS ONE

Dear Dr. Chen,

Thank you for submitting your manuscript to PLOS ONE. After careful consideration, we feel that it has merit but does not fully meet PLOS ONE’s publication criteria as it currently stands. Therefore, we invite you to submit a revised version of the manuscript that addresses the points raised during the review process.

The reviewers have commented on your manuscript. They are asking to revise the manuscript.

We look forward to receiving your revised manuscript.

Kind regards,

Vedik Basetti, Ph.D

Academic Editor

PLOS ONE

4. Thank you for stating the following financial disclosure: [This study was supported by the Central Government Guiding Local Science and Technology Development Special Fund Project "Application and Promotion of Key Technologies for Integrated Water and Fertilizer Management in Desert Wheat" (ZYYD2024CG19); the Ministry of Science and Technology Innovation 2030 Major Project on "New Generation Artificial Intelligence"—"Integrated Application and Demonstration of Key Technologies for Smart Farms" (2022ZD0115805); and the Xinjiang Uygur Autonomous Region Major Science and Technology Special Project "Research on Intelligent Control System for Integrated Water and Fertilizer Management in Farmland" (2022A02011-3).]. Please state what role the funders took in the study. If the funders had no role, please state: "The funders had no role in study design, data collection and analysis, decision to publish, or preparation of the manuscript." If this statement is not correct you must amend it as needed. Please include this amended Role of Funder statement in your cover letter; we will change the online submission form on your behalf.

5. Please note that your Data Availability Statement is currently missing [the repository name and/or the DOI/accession number of each dataset OR a direct link to access each database]. If your manuscript is accepted for publication, you will be asked to provide these details on a very short timeline. We therefore suggest that you provide this information now, though we will not hold up the peer review process if you are unable.

6. PLOS requires an ORCID iD for the corresponding author in Editorial Manager on papers submitted after December 6th, 2016. Please ensure that you have an ORCID iD and that it is validated in Editorial Manager. To do this, go to ‘Update my Information’ (in the upper left-hand corner of the main menu), and click on the Fetch/Validate link next to the ORCID field. This will take you to the ORCID site and allow you to create a new iD or authenticate a pre-existing iD in Editorial Manager.

7. "Please ensure that you refer to Figure 1 in your text as, if accepted, production will need this reference to link the reader to the figure.

Additional Editor Comments (if provided):

Reviewers' comments:

Reviewer's Responses to Questions

**Comments to the Author**

1. Is the manuscript technically sound, and do the data support the conclusions?

Reviewer #1: Yes

Reviewer #2: Partly

Reviewer #3: Yes

2. Has the statistical analysis been performed appropriately and rigorously? 

Reviewer #1: Yes

Reviewer #2: Yes

Reviewer #3: Yes

3. Have the authors made all data underlying the findings in their manuscript fully available?

Reviewer #1: Yes

Reviewer #2: Yes

Reviewer #3: Yes

4. Is the manuscript presented in an intelligible fashion and written in standard English?

Reviewer #1: No

Reviewer #2: Yes

Reviewer #3: Yes

5. Review Comments to the Author

Reviewer #1: The article titled “Multi-objective Optimization Method for Path Planning and

Task Allocation in Farmland Land Levelling Based on an Im-proved NSGA-III”, relates to my area of interest that’s why I recommend some points which may help in order to improve the readability as well as overall structure of this manuscript. The following are my suggestions, recommendations and questions for this article which may help to improve the quality of this manuscript are as follows.

1. Title

- What’s the novelty of this article

- Need to summarise the abstract

2. Abstract

• Background of your proposed framework may be more elaborated

• Must use the contrast word i.e., on one hand and on the other hand.

• By using the proposed system/model/control technique, how can you relate that your work is efficient than the other techniques must add contrast in it.

3. Proper Structure is required

• Introduction must be divided in to four sections 1. Motivation 2. Related Work 3. Your contributions 4. Organisation of this manuscript

• After introduction must add another heading Problem statement and its proposed solution

4. Introduction (revision is required in this section)

• Background of the area may be elaborated more

• Put some pictures related to the area of study

• Problem statement and solution are missing

5. Major Grammar Issues

Emphasising clarity

• Reducing reliance on pronouns and determiners, such as ’this’, ’that’, ’it’, ’their’, etc., improves the readability of an academic paper.

Depersonalising sentences

• In academic writing, instead of using first-person pronouns, such as ’I’ and ’We’, focusing on actions is preferable. Please note the following examples.

Lexical Issues

• 2.1. ’such as’ instead of ’like’

• Correct use of ’as’

• ’complete’ instead of ’whole’

Articles

• Mastering articles can be a challenge for non-native speakers, but Figure 1 offers a helpful guide (Please see Schrampfer (2016) for more detail).

• Example

Xyz denotes operational expenses directly related to the controller operation. ✗

xyz denotes operational expenses directly related to controller operation. ✓

6. System model (revision is required in this section)

• Describe your proposed model "sufficiently" detailed such that others can redo your experiment(s).

• Mathematical equations are not written appropriately. It must be simplified

• Big equations i.e., must be written in a curl bracket

• Recheck equation (13), it has issues

7. Algorithm

- Need to write in a proper way

8. Results and Discussions

• Comparison with some other experimental results are missing.

• Validation of the results is missing

• Need to compare with at-least two methods one is traditional and add another state of the art method in order to justify your proposed method.

9. Figures & tables

• Need to add more figures as “a picture is worth a thousand words?”

Figure 1 need to remake

• Visibility of figures are not appropriate use a proper software like “inkscape” or so on.

• Method figure is worst need to remake

• Figure 4: Single-field path planning flow chart why make single field ? multi is best option here in order to validate the work

10. References

Check all the references carefully. Need to add more references. Some of the related reference written below may be cited and added in this section to further strengthen this work.

https://doi.org/10.3390/s21113820

10.1108/IJICC-02-2014-0005

General Comments

• Grammar is weak need to rewrite most the parts

• Background of the proposed area may elaborated more.

• Some observations in the introduction and system model part as I reflect it on point number 3 & 4 so kindly incorporate all those points accordingly.

• Result part must also be addressed as I mentioned in point number 5.

• Check all the references carefully and incorporate the above mentioned paper in your reference list.

• Must add multi UAV’s concept in this manuscript its missing

• Problem statement and its solution must be defined

Reviewer #2: 1. Please explain why the WOA algorithm is chosen in this paper. Specifically, it is recommended to discuss the advantages of this algorithm in solving certain problems and compare it with other optimization algorithms.

2. Please provide a comprehensive explanation of the rationale behind the assertion that six coordinates can accurately represent the boundaries of a parcel of land. This may necessitate an explication of geometric properties, the distribution of points, and their respective positions in space.

3. It is recommended that the authors provide a comprehensive explication of the denotation of each letter in Figure 3, as well as in other figures and equations.

4. The existing description of the fine-tuning stage of the BWO algorithm lacks sufficient detail. It is recommended that further elaboration be provided on this optimization stage, with a clear delineation of the specific processes and purposes of each step.

5. The unit of measurement is missing in Figure 12-15. The relevant unit information needs to be supplemented to improve the clarity and understandability of the figures.

Reviewer #3: 1 The main motivations of the proposed algorithm are not clear. The advantages and disadvantages of previous work should be discussed more comprehensively and the effectiveness of the proposed algorithm for solving existing problems should be emphasized.

2 The existing approaches should be classified into different types, and the advantages and disadvantages of each type should be clearly introduced.

3 There is not a part to express the novelty in the whole of the article. The reviewer cannot find whether the idea proposed in this paper is better than those in other papers. Please point out the novelty of the paper in the “Introduction” section.

4 The related work does not cover many contributions on path or mission planning. For example, Coverage Path Planning of Heterogeneous Unmanned Aerial Vehicles Based on Ant Colony System, An adaptive clustering-based algorithm for automatic path planning of heterogeneous UAVs, Global-and-Local Attention-Based Reinforcement Learning for Cooperative Behaviour Control of Multiple UAVs. All these works focus on the path or mission planning problem. It is suggested to cite the above article and analyze the differences.

5 In Section 4, description about the experiments is not perfect. Some experiment parameters should be introduced in this section such the readers know how to repeat the experiments and evaluate the experiment results. For example, how many times was each experiment run? Is each point in the tables represent the average value of several experiment results? Please add more sentences to introduce these parameters in the revised manuscript.

6. PLOS authors have the option to publish the peer review history of their article (what does this mean?). If published, this will include your full peer review and any attached files.

Reviewer #1: No

Reviewer #2: No

Reviewer #3: No

---

## [Author Response · Author response to Decision Letter 0]

20 Nov 2024

Dear Editor and Reviewers,

Thank you for your detailed review and valuable suggestions regarding our manuscript. We have carefully considered all the comments and made the necessary revisions to the manuscript. Below is our point-by-point response to each of the reviewers' comments and the specific changes made:

1. Regarding manuscript format:

The paper has been reformatted according to the journal's template, and we have verified that all figures and tables are correctly cited in the text.

2. Regarding permits and permissions:

In the “3. Farmland environment modeling” section, we have added the following statement: "The field studies were conducted on private agricultural land with explicit permission from the landowners. No specific permits were required as the research involved standard farming practices and did not involve protected species or conservation areas."

3. Regarding the role of funders:

We have added the following statement to the Cover Letter:

"The funders provided specific support for this research:

The Central Government Guiding Local Science and Technology Development Special Fund Project (ZYYD2024CG19) covered publication fees and travel expenses for field surveys and data collection.

The Ministry of Science and Technology Innovation 2030 Major Project (2022ZD0115805) supported the purchase of UAV and marking equipment for experimental data collection.

The Xinjiang Uygur Autonomous Region Major Science and Technology Special Project (2022A02011-3) provided insurance coverage and living allowances for research personnel.

Reviewer 1:

1: The article titled “Multi-objective Optimization Method for Path Planning and

Task Allocation in Farmland Land Levelling Based on an Im-proved NSGA-III”, relates to my area of interest that’s why I recommend some points which may help in order to improve the readability as well as overall structure of this manuscript. The following are my suggestions, recommendations and questions for this article which may help to improve the quality of this manuscript are as follows

1. Title

- What’s the novelty of this article

- Need to summarise the abstract

Response: The original title ' Multi-objective Optimization Method for Path Planning and Task Allocation in Farmland Land Levelling Based on an Improved NSGA-III' has been revised to ' A two-stage hybrid NSGA-III with BWO for path planning and task allocation in agricultural land preparation'. The new title better reflects the paper's innovations and research methodology.

2. Abstract

• Background of your proposed framework may be more elaborated

• Must use the contrast word i.e., on one hand and on the other hand.

• By using the proposed system/model/control technique, how can you relate that your work is efficient than the other techniques must add contrast in it.

Response: The original background statement merely mentioned 'to improve agricultural production efficiency and reduce operational costs.' This has been expanded to: 'Large-scale automated farmland preparation operations face challenges in path planning efficiency and resource allocation imbalance. To improve agricultural production efficiency and reduce operational costs, this study proposes an improved land preparation path planning method.' Additional background information is elaborated in the introduction.

The experimental results have been enhanced with comparative data: For single-field operations, compared to random operation directions, path length decreased by 1.9%-3.1%, turning frequency reduced by 19.5%-24.0%, and coverage increased by 1.0%-1.4%. In multi-machine scheduling, compared to NSGA-II, NSGA-III, and MOEA/D algorithms, BNSGA-III demonstrated improvements of 12.3%-34.4% in total travel distance, 60.9%-66.2% in path balance, and 78.7%-92.9% in workload distribution. The algorithm's superiority is further validated through key performance indicators including convergence (IGD), solution quality (HV), and diversity (Spread), confirming its excellence in solution quality, convergence, and diversity aspects.

3. Proper Structure is required

• Introduction must be divided in to four sections 1. Motivation 2. Related Work 3. Your contributions 4. Organisation of this manuscript

• After introduction must add another heading Problem statement and its proposed solution

Response: Following the suggestions, the paper's structure has been reorganized. The introduction has been divided into four sections: 1. Motivation 2. Related Work 3. Contributions 4. Paper Organization.

Section 1.1 (Research Motivation and Background) has been enhanced with agricultural automation background, the importance of land preparation operations, key technical challenges, and supporting statistical data.

Section 1.2 (Related Work) systematically reviews previous research, including 1.2.1 single-field path planning and 1.2.2 multi-machine collaborative scheduling, followed by 1.2.3 identifying limitations in existing methods.

Section 1.3 (Contributions) describes algorithmic innovations, particularly the improved BNSGA-III algorithm and comprehensive hierarchical optimization framework.

Section 1.4 (Paper Organization) clearly outlines the content and connections between chapters.

A new 'Problem Statement and Solution Framework' section has been added as Section 2."

4. Introduction (revision is required in this section)

• Background of the area may be elaborated more

• Put some pictures related to the area of study

• Problem statement and solution are missing

Response: The background section has been expanded, and a new agricultural path planning and task allocation diagram (Figure 1) has been added to visually demonstrate the problem. Section 2 has been supplemented with problem statements and proposed solutions.

5. Major Grammar Issues

Emphasising clarity

• Reducing reliance on pronouns and determiners, such as ’this’, ’that’, ’it’, ’their’, etc., improves the readability of an academic paper.

Depersonalising sentences

• In academic writing, instead of using first-person pronouns, such as ’I’ and ’We’, focusing on actions is preferable. Please note the following examples.

Lexical Issues

• 2.1. ’such as’ instead of ’like’

• Correct use of ’as’

• ’complete’ instead of ’whole’

Articles

• Mastering articles can be a challenge for non-native speakers, but Figure 1 offers a helpful guide (Please see Schrampfer (2016) for more detail).

• Example

Xyz denotes operational expenses directly related to the controller operation. ✗

xyz denotes operational expenses directly related to controller operation. ✓

Response: The entire manuscript has been thoroughly reviewed and revised to reduce dependency on pronouns and determiners, enhance grammar, and improve academic readability. If there are any other language issues, we will continue to refine them.

6. System model (revision is required in this section)

• Describe your proposed model "sufficiently" detailed such that others can redo your experiment(s).

• Mathematical equations are not written appropriately. It must be simplified

• Big equations i.e., must be written in a curl bracket

• Recheck equation (13), it has issues

Response: Problem modeling has been added, and the model section now includes detailed pseudocode and flowcharts to clarify algorithm implementation details. Experimental parameters have been supplemented to ensure reproducibility. Mathematical equations have been reviewed and revised for clarity and standardization, including the correction of equation (13).

7. Algorithm

- Need to write in a proper way

Response: The algorithm section has been rewritten with the addition of formulas and pseudocode to provide more detailed algorithm descriptions.

8. Results and Discussions

• Comparison with some other experimental results are missing.

• Validation of the results is missing

• Need to compare with at-least two methods one is traditional and add another state of the art method in order to justify your proposed method.

Response: The results and discussion section has been enhanced with comparisons to NSGA-II, NSGA-III, and MOEA/D algorithms. The improvements show total distance reduction of 20.3%, 14.1%, and 34.4%, transfer distance reduction of 66.2%, 60.9%, and 66.2%, and workload balance improvement of 90.2%, 78.7%, and 92.9% respectively. The iteration curves demonstrate faster convergence of the proposed algorithm, while comparisons of HV, IGD, and Spread metrics (Figures 12-14) validate its comprehensive advantages in solution quality, convergence, and diversity.

9. Figures & tables

• Need to add more figures as “a picture is worth a thousand words?”

Figure 1 need to remake

• Visibility of figures are not appropriate use a proper software like “inkscape” or so on.

• Method figure is worst need to remake

• Figure 4: Single-field path planning flow chart why make single field ? multi is best option here in order to validate the work

Response: Indeed, Inkscape is an excellent and free illustration software. We have added Figure 1 showing the farmland operation scheduling scheme, which illustrates path planning and task allocation strategies and demonstrates the complete workflow. Figure 2 has been added to show the technical framework of agricultural machinery path planning and task allocation, clearly displaying the three-layer architecture and module relationships. Figure 1 (now Figure 3) has been redesigned to improve visibility, and Figure 4 (now Figure 2) has been modified accordingly.

10. References

Check all the references carefully. Need to add more references. Some of the related reference written below may be cited and added in this section to further strengthen this work.

https://doi.org/10.3390/s21113820

10.1108/IJICC-02-2014-0005

Response: All references have been carefully checked to ensure correct citations and standardized formatting. The recommended references have been incorporated to enrich the manuscript.

General Comments

• Grammar is weak need to rewrite most the parts

• Background of the proposed area may elaborated more.

• Some observations in the introduction and system model part as I reflect it on point number 3 & 4 so kindly incorporate all those points accordingly.

• Result part must also be addressed as I mentioned in point number 5.

• Check all the references carefully and incorporate the above mentioned paper in your reference list.

• Must add multi UAV’s concept in this manuscript its missing

• Problem statement and its solution must be defined

Response: Thank you for your suggestions. Comprehensive revisions have been made:

1. Grammar and Language: The entire manuscript has been reviewed for academic language standardization, removing subjective expressions and optimizing sentence structures.

2. Research Background: The background section has been expanded with additional data support and clear technical challenges.

3. Introduction and System Model: The introduction has been reorganized into four clear sections, with additional references enriching the content. A new 'Problem Statement and Solution Framework' section has been added, and the system model's mathematical description has been improved with the addition of technical framework diagrams.

4. Results Section: Comparisons with both traditional and state-of-the-art methods have been added, supplemented with detailed performance evaluation data and enhanced result validation.

5. References: Citation standards have been improved, and the recommended papers have been incorporated into the reference list. The multi-UAV coordination concept has been added.

Reviewer 2:

1. Please explain why the BWO algorithm is chosen in this paper. Specifically, it is recommended to discuss the advantages of this algorithm in solving certain problems and compare it with other optimization algorithms.

Response: The BWO algorithm offers several distinct advantages: Its three-stage search mechanism (exploration, exploitation, and whale fall) achieves balance between global and local search; adaptive parameter adjustment enhances algorithmic adaptability; and parallel search strategies help escape local optima. In multi-objective optimization problems, the algorithm effectively handles objective conflicts, maintains solution diversity, and prevents premature convergence. However, BWO also has limitations, primarily in early-stage search efficiency and lack of problem-specific optimization mechanisms. To address these limitations, this paper combines BWO with NSGA-III, utilizing NSGA-III's reference point mechanism and non-dominated sorting strategy to enhance performance in multi-objective optimization while leveraging BWO's three-stage search mechanism to strengthen global search capabilities.

2. Please provide a comprehensive explanation of the rationale behind the assertion that six coordinates can accurately represent the boundaries of a parcel of land. This may necessitate an explication of geometric properties, the distribution of points, and their respective positions in space.

Response: We apologize for the unclear presentation. In the high-precision mapping using UAVs, each field boundary contains over 120 coordinate points due to detailed collection and large field boundaries. The table shows only six coordinates per field due to space limitations. We will revise the manuscript to clarify this and adjust the data presentation format for better clarity.

3. It is recommended that the authors provide a comprehensive explication of the denotation of each letter in Figure 3, as well as in other figures and equations.

Response: All letter denotations in Figure 3 and other figures have been supplemented with clear explanations of their physical meanings.

4. The existing description of the fine-tuning stage of the BWO algorithm lacks sufficient detail. It is recommended that further elaboration be provided on this optimization stage, with a clear delineation of the specific processes and purposes of each step.

Response: Section 3.2.3 has been enhanced with algorithm pseudocode providing detailed descriptions that clearly demonstrate the implementation process of each algorithm stage.

5. The unit of measurement is missing in Figure 12-15. The relevant unit information needs to be supplemented to improve the clarity and understandability of the figures.

Response: Units of measurement have been added to all figures. In Figure 3, the following notations have been clarified: ABCD represents turning trajectories, W denotes operating width (implement working width), r indicates minimum turning radius, R represents the actual turning radius of agricultural machinery, θ shows the operation direction angle with field boundary, Ci denotes the cost of the i-th turning method (i=1,2,3,4), and Li represents the headland width reserved for each turning method.

Reviewer 3:

1 The main motivations of the proposed algorithm are not clear. The advantages and disadvantages of previous work should be discussed more comprehensively and the effectiveness of the proposed algorithm for solving existing problems should be emphasized.

Response: The abstract and introduction have been revised to discuss the advantages and disadvantages of previous work. Given that multi-machine scheduling involves large decision spaces with multiple objective functions and complex constraints, single algorithms struggle to balance global and local optimization. BWO's three-stage mechanism effectively balances global and local search, while NSGA-III's reference point mechanism ensures solution distribution in high-dimensional problems. The combination of these algorithms complements their respective limitations. Therefore, this paper proposes an effective integration of BWO and NSGA-III, using multi-objective optimization to balance various indicators, adaptive mechanisms to handle complex constraints, improved path planning efficiency, and optimized resource allocation to solve large-scale agricultural machinery scheduling problems.

2 The existing approaches should be classified into different types, and the advantages and disadvantages of each type should be clearly introduced. 

Response: The introduction has been revised to classify problems into single-field and multi-field categories. Previous methods are analy

---

## [Decision Letter · Decision Letter 1]

29 Nov 2024

A two-stage hybrid NSGA-III with BWO for path planning and task allocation in agricultural land preparation

PONE-D-24-46846R1

Dear Dr. Chen,

We’re pleased to inform you that your manuscript has been judged scientifically suitable for publication and will be formally accepted for publication once it meets all outstanding technical requirements.

Kind regards,

Vedik Basetti, Ph.D

Academic Editor

PLOS ONE

Additional Editor Comments (optional):

Reviewers' comments:

Reviewer's Responses to Questions

**Comments to the Author**

1. If the authors have adequately addressed your comments raised in a previous round of review and you feel that this manuscript is now acceptable for publication, you may indicate that here to bypass the “Comments to the Author” section, enter your conflict of interest statement in the “Confidential to Editor” section, and submit your "Accept" recommendation.

Reviewer #1: All comments have been addressed

Reviewer #2: All comments have been addressed

2. Is the manuscript technically sound, and do the data support the conclusions?

Reviewer #1: Yes

Reviewer #2: Yes

3. Has the statistical analysis been performed appropriately and rigorously? 

Reviewer #1: Yes

Reviewer #2: Yes

4. Have the authors made all data underlying the findings in their manuscript fully available?

Reviewer #1: Yes

Reviewer #2: Yes

5. Is the manuscript presented in an intelligible fashion and written in standard English?

Reviewer #1: Yes

Reviewer #2: Yes

6. Review Comments to the Author

Reviewer #1: All my concern is already addressed by the authors, now I recommend to accept this article in its present from.

Reviewer #2: It is recommended to carefully check the information in each reference before submitting the final draft to ensure that the information is complete and in a uniform format.

7. PLOS authors have the option to publish the peer review history of their article (what does this mean?). If published, this will include your full peer review and any attached files.

Reviewer #1: No

Reviewer #2: No

---

## [Editor Report · Acceptance letter]

13 Dec 2024

PONE-D-24-46846R1 

PLOS ONE

Dear Dr. Chen, 

I'm pleased to inform you that your manuscript has been deemed suitable for publication in PLOS ONE. Congratulations! Your manuscript is now being handed over to our production team.

Kind regards, 

on behalf of

Dr. Vedik Basetti 

Academic Editor

PLOS ONE